# mTORC1-induced retinal progenitor cell overproliferation leads to accelerated mitotic aging and degeneration of descendent Müller glia

**Soyeon Lim[1], You-Joung Kim[1], Sooyeon Park[1], Ji-heon Choi[1], Young Hoon Sung[2,3], Katsuhiko Nishimori[4], Zbynek Kozmik[5], Han-Woong Lee[2], Jin Woo Kim[1]\***

[1]Department of Biological Sciences, Korea Advanced Institute of Science and Technology, Daejeon, Republic of Korea; [2]Department of Biochemistry, College of Life Science and Biotechnology, Yonsei University, Yonsei, Republic of Korea; [3]Department of Convergence Medicine, Asan Medical Center, University of Ulsan College of Medicine, Seoul, Republic of Korea; [4]Department of Obesity and Internal Inflammation; Bioregulation and Pharmacological Medicine, Fukushima Medical University, Fukushima, Japan; [5]Institute of Molecular Genetics of the Czech Academy of Sciences, Prague, Czech Republic

**Abstracts** Retinal progenitor cells (RPCs) divide in limited numbers to generate the cells comprising vertebrate retina. The molecular mechanism that leads RPC to the division limit, however, remains elusive. Here, we find that the hyperactivation of mechanistic target of rapamycin complex 1 (mTORC1) in an RPC subset by deletion of *tuberous sclerosis complex 1* (*Tsc1*) makes the RPCs arrive at the division limit precociously and produce Müller glia (MG) that degenerate from senescence-associated cell death. We further show the hyperproliferation of *Tsc1*-deficient RPCs and the degeneration of MG in the mouse retina disappear by concomitant deletion of *hypoxia-induced factor 1-alpha* (*Hif1a*), which induces glycolytic gene expression to support mTORC1-induced RPC proliferation. Collectively, our results suggest that, by having mTORC1 constitutively active, an RPC divides and exhausts mitotic capacity faster than neighboring RPCs, and thus produces retinal cells that degenerate with aging-related changes.

## Editor's evaluation

Using a broad range of genetic and biochemical strategies, this study shows that hyperactivation of mTOR in retinal progenitors induces hyperproliferation and thus prematurely exhaust their mitotic capacity. They also show that these effects are related to Hif1alpha activation and metabolism. The study will be of considerably interest beyond field of retinal development, as it clearly links cell metabolism to tissue growth and perhaps competition.

## Introduction

Neural progenitor cells (NPCs) divide repeatedly during development to generate the cells of vertebrate neural tissues (*Homem et al., 2015*; *Obernier and Alvarez-Buylla, 2019*). NPCs are present for a limited period before entering a final cell division for the differentiation to neurons and glia. Consequently, NPCs are not seen in the majority of adult neural tissues, except for sub-brain areas that exhibit continued neurogenesis.

**\*For correspondence:** jinwookim@kaist.ac.kr

**Competing interest:** The authors declare that no competing interests exist.

The factors that determine when an NPC enters the final division and how long it keeps division capacity, however, still remain elusive. It has been identified that the mitotic characteristics of NPCs change continually in the development of neural tissues. Most NPCs divide symmetrically to expand themselves during early development; thereafter, the asymmetrically dividing NPCs increase to preserve the NPC population while actively generating the various types of neurons and glia (*Homem et al., 2015*; *Obernier and Alvarez-Buylla, 2019*). The length of the NPC cell cycle also increases with development, while the NPC division capacity decreases (*Alexiades and Cepko, 1996*; *Ohnuma and Harris, 2003*). Given the heterogeneity of the NPC division mode and cell cycle length, it is believed that cumulative division numbers might be diversified among NPCs even within the same neural tissues in development. Consequently, some NPCs might have already reached their mitotic division limits while neighboring NPCs are still able to divide further. However, the division capacity of an NPC in developing neural tissue has not been empirically quantified to date.

The mouse retina has been used as a model system in studies aiming to identify general features of mammalian NPC proliferation and differentiation (*Cepko, 2014*). Two retinal progenitor cell (RPC) populations have been identified in mouse retina (*Clark et al., 2019*). The early RPC population produces retinal ganglion cells (RGCs), amacrine cells (ACs), horizontal cells (HCs), cone photoreceptors (cPRs), and some rod photoreceptors (rPRs), whereas the late RPC population mainly produces bipolar cells (BCs), Müller glia (MG), and some rPRs (*Cepko, 2014*; *Clark et al., 2019*). Given the temporal-specific development of retinal cell types, cumulative division numbers of RPCs producing MG, which is the last-born retinal cell type, are likely different from those of RPCs generating RGCs, which is the first-born retinal cell type.

The decision of an RPC to exit the cell cycle and undergo differentiation is regulated by various factors. For example, RPCs extended division capacity in mice deficient of the *cell cycle-dependent kinase inhibitor 1b* (*Cdkn1b/p27*) (*Levine et al., 2000*), whereas they prematurely exited the cell cycle and were extinguished precociously in mice lacking *cyclin D1* (*Ccnd1*) (*Das et al., 2012*). Consequently, the production of MG is decreased in *Ccnd1*-deficient mouse retina, whereas it is extended in *Cdkn1*-deficient mouse retina. Therefore, cumulative division numbers of *Cdkn1*-deficient RPCs are likely to be greater than those of *Ccnd1*-deficient RPCs when these cells undertake the production of MG in the postnatal retina.

The speed of the cell cycle could also affect the cumulative division number of RPCs as well. *Tuberous sclerosis complex 1* (*Tsc1*)-deficient mouse RPCs, which have hyperactive mechanistic target of rapamycin complex 1 (mTORC1), were found to complete a cell cycle more quickly than wild-type mouse RPCs (*Choi et al., 2018*). This resulted in faster accumulation of newborn cells in the *Tsc1*-deficient mouse retinas than in wild-type retinas. Here, we also found faster accumulation of cells in the mouse retina where *Tsc1* was deleted in the minor RPC population derived from the ciliary margin (CM). This made the CM RPC-derived *Tsc1*-deficient clones invade into neighboring areas, where wild-type clones grow slowly, and occupy almost entire territory of the mature mouse retina. However, later, the retinal cells, especially the MG, produced from *Tsc1*-deficient CM RPCs take on senescent characteristics and degenerate to form rosette structures in the retinas. The *Tsc1*-deficient retinal cells were found to be mitotically older than those derived from neighboring wild-type RPC clones. The precocious aging phenotypes of the *Tsc1* conditional knock-out (cko) mouse retina were rescued by concomitant deletion of *hypoxia-induced factor 1-alpha* (*Hif1a*), which supports RPC proliferation by inducing the expression of glycolytic enzymes that can supply ATP in *Tsc1*-deficient RPCs. These results suggest that there are limits to RPC mitotic division that can be reached precociously by hyper-expansion of an RPC clone in the developing retina.

## Results

### Degeneration of MG derived from *Tsc1*-deficient CM RPCs

mTORC1 activity, which leads to the phosphorylation of ribosomal protein S6 (pS6) via the activation of S6 kinase 1 (S6K1), was detectable in developing mouse retina but was absent in the neighboring retinal pigment epithelium (RPE) and CM (*Figure 1—figure supplement 1*). However, it is unknown why mTORC1 activity is diversified in the optic neuroepithelial continuum of the retina-CM-RPE, which were shown to exhibit differential proliferation rates (*Moon et al., 2018*). To understand the physiological importance of the spatially differentiated mTORC1 activation, we ectopically increased

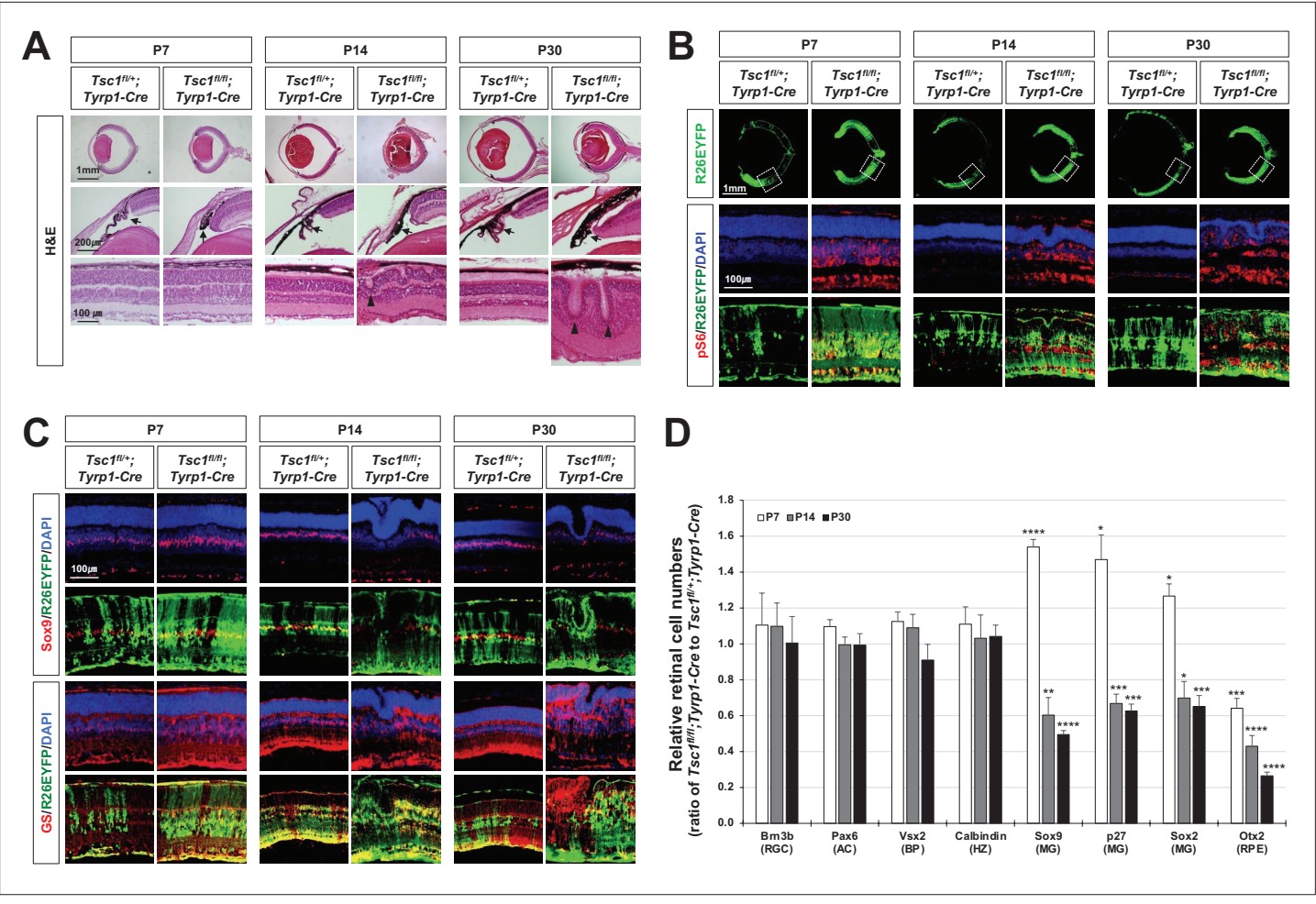

**Figure 1.** MG degeneration and rosette formation in *Tsc1fl/fl;Tyrp1-Cre* mouse retina. (**A**) Developmental changes of eye and retinal structures in *Tsc1fl/+;Tyrp1-Cre* and *Tsc1fl/fl;Tyrp1-Cre* littermate mice were examined by hematoxylin and eosin (H&E) staining of the eye sections. ONL, outer nuclear layer; OPL, outer plexiform layer; INL, inner nuclear layer; IPL, inner plexiform layer; GCL, ganglion cell layer. (**B**) The Cre-affected cells in the mouse retinas were visualized by R26EYFP Cre reporter, and mTORC1 activation of the cells was determined by immunostaining of pS6. Nuclei of the cells were visualized by DAPI staining. (**C**) Distributions of MG in the mouse retinas were examined by immunostaining of the MG markers, Sox9 and glutamine synthetase (GS). (**D**) Retinal cell type-specific marker-positive cells among DAPI-positive total retinal cells in 350 μm × 350 μm areas were counted and shown their relative values against those of *Tsc1fl/+;Tyrp1-Cre* mouse retinas in the graph. Representative staining images of retinal markers are provided in (**C**) and *Figure 1—figure supplement 2A*. Error bars denote standard deviations (SD). The numbers of samples are 5 from five independent litters. **p < 0.01; ***p < 0.001; ****p < 0.0001.

The online version of this article includes the following figure supplement(s) for figure 1:

**Figure supplement 1.** Spatially restricted mTORC1 activation in developing mouse eyes.

**Figure supplement 2.** Cell composition of *Tsc1fl/+;Tyrp1-Cre* and *Tsc1fl/fl;Tyrp1-Cre* mouse retinas.

**Figure supplement 3.** Apoptotic cell death of *tuberous sclerosis complex 1 (Tsc1)*-deficient MG.

**Figure supplement 4.** MG degeneration and rosette formation in *Tsc2fl/fl;Tyrp1-Cre* mouse retinas.

mTORC1 activity in the RPE and CM by deleting *Tsc1*, which is a negative upstream regulator of mTORC1 (*Gao et al., 2002*; *Saxton and Sabatini, 2017*). To this end, we bred *Tsc1flox/flox* (*Tsc1fl/fl*) mice with *tyrosinase-related protein 1-Cre* (*Tyrp1-Cre*) mice, which express Cre recombinase in both the RPE and the CM cells that have a potential to become RPCs in the peripheral retina (*Fischer et al., 2013*; *Mori et al., 2002*; *Figure 1A*). As expected, Cre-affected cells, which were visualized by a fluorescent Cre reporter ROSA26EYFP (R26EYFP), were detectable in the RPE, CM, and peripheral retina of *Tsc1fl/+;Tyrp1-Cre* mice (*Figure 1B*, top row). The R26EYFP-positive cells were found farther in the central part of *Tsc1fl/fl;Tyrp1-Cre* mouse retinas, suggesting that the CM-derived *Tsc1*-deficient cell populations were expanded more centrally in the retinas.

We found that the eyes of *Tsc1$^{fl/fl}$;Tyrp1-Cre* mice at P14 and P30 were significantly smaller than those of *Tsc1$^{fl/+}$;Tyrp1-Cre* littermates (**Figure 1A and B**, top rows), and their ciliary body (CB) and iris were malformed (arrows in **Figure 1A**, middle row). Moreover, the *Tsc1$^{fl/fl}$;Tyrp1-Cre* mouse retinas exhibited multiple rosette structures (arrowheads in **Figure 1A**, bottom row). However, these phenotypes were not observed in P7 *Tsc1$^{fl/+}$;Tyrp1-Cre* and *Tsc1$^{fl/fl}$;Tyrp1-Cre* littermate mouse eyes (**Figure 1A and B**, two leftmost columns). The results suggest that the structural alterations in *Tsc1$^{fl/fl}$;Tyrp1-Cre* mouse retinas began after the first postnatal week.

It is known that the loss of MG is a major cause of retinal rosette formation (**Willbold et al., 2000**). We found that the cells expressing the MG markers, including p27, SRY-box transcription factors 2 and 9 (Sox2 and Sox9), and glutamine synthetase, were decreased significantly in P14 and P30 *Tsc1$^{fl/fl}$;Tyrp1-Cre* mouse retinas compared with *Tsc1$^{fl/+}$;Tyrp1-Cre* littermate retinas, whereas the numbers of other retinal cell types were not significantly different in those two retinas (**Figure 1C and D**; **Figure 1—figure supplement 2A and B**). However, the numbers of MG were rather increased in P7 *Tsc1$^{fl/fl}$;Tyrp1-Cre* mouse retina (**Figure 1D**; **Figure 1—figure supplement 2B**); this is likely due to the developmental acceleration of MG production, as it was reported previously (**Choi et al., 2018**). The numbers of apoptotic cells, which were assessed by TUNEL (terminal deoxynucleotidyl transferase dUTP nick end labeling) and immunostaining of cleaved active caspase-3 (Casp-3), were also elevated in the MG population of *Tsc1$^{fl/fl}$;Tyrp1-Cre* mouse retinas (**Figure 1—figure supplement 3A,B,C,D,E,F,G**). Furthermore, the apoptotic cells were enriched in the *Tsc1*-deficient cell population positive for the Cre reporter, ROSA26$^{tdTomato}$ (R26tdTom) (**Figure 1—figure supplement 3A and C**), suggesting that *Tsc1* deletion had autonomous effects on MG degeneration. These findings suggest that the number of MG in *Tsc1$^{fl/fl}$;Tyrp1-Cre* mice might decrease below a critical level needed to maintain an intact retinal structure, due to the enhanced degeneration of MG.

Retinal rosettes and degeneration of MG were also observed in *Tsc2$^{fl/fl}$;Tyrp1-Cre* mice, in which the other TSC component, *Tsc2* (**Inoki et al., 2002**; **Saxton and Sabatini, 2017**), was deleted in the RPE and CM cells (**Figure 1—figure supplement 4A and C**). However, eyes of *Tsc2$^{fl/fl}$;Tyrp1-Cre* mice were not significantly different in size compared to those of *Tsc2$^{fl/+}$;Tyrp1-Cre* mice, and their CB and iris were intact (**Figure 1—figure supplement 4A and B**). These results suggest that the microphthalmia and CB/iris malformation of *Tsc1$^{fl/fl}$;Tyrp1-Cre* mice is not due to mTORC1 hyperactivation; instead, they might be caused by cellular events that are regulated by Tsc1.

## MG degeneration and retinal rosette formation are not caused by the hyperactivation of mTORC1 in mature retina

To examine whether the MG degeneration and retinal rosette formation were due to mTORC1 hyperactivation in MG, we deleted *Tsc1* using the MG-specific *solute carrier family 1 member 3-CreERT2* (*Slc1a3-CreERT2*), which is active in the presence of estrogen analog, tamoxifen (Tam) (**de Melo et al., 2012**). However, we could not observe MG degeneration or retinal rosettes in P30 *Tsc1$^{fl/fl}$;Slc1a3-CreERT2* mice, from which *Tsc1* was deleted in MG beginning at P10 by Tam injection (**Figure 2A**).

We also found that the retinal rosettes of *Tsc1$^{fl/fl}$;Tyrp1-Cre* mice were still evident when mTORC1 was inhibited by daily injection of a chemical inhibitor, rapamycin, during the second postnatal week (**Figure 2B**). The numbers of MG in *Tsc1$^{fl/fl}$;Tyrp1-Cre* mouse retinas were also not recovered by rapamycin treatment, whereas pS6 was completely absent from the rapamycin-treated mouse retinas. Collectively, these results suggest that the degenerative phenotypes are not due to mTORC1 activation in the MG; instead, they likely arise from mTORC1 activation in the developing mouse retina.

## MG degeneration and retinal rosette formation cannot be induced by *Tsc1* deletion in the majority of RPC and RPE populations

Interestingly, in contrast to *Tsc1$^{fl/fl}$;Tyrp1-Cre* mice, retinal rosettes were not seen in *Tsc1$^{fl/fl}$;Chx10-Cre* and *Tsc1$^{fl/fl}$;Mlana-Cre* mice (**Figure 3A**, third and fifth columns from left), in which *Tsc1* was deleted from most of the RPCs by *Chx10-EGFP/cre* (*Chx10-Cre*) and from the RPE by *Mlana-Cre*, respectively (**Aydin and Beermann, 2011**; **Rowan and Cepko, 2004**). The embryonic and early postnatal mouse RPE was shown to make direct contact with adjacent RPCs to regulate neurogenic capacity of the RPCs (**Ha et al., 2017**). Therefore, we speculated that the phenotypes of *Tsc1$^{fl/fl}$;Tyrp1-Cre* mice might be induced only when *Tsc1* was lost commonly in the RPE and RPC. However, we failed to find the decrease of MG as well as retinal rosettes in *Tsc1$^{fl/fl}$;Chx10-Cre;Mlana-Cre* mouse eyes, in which *Tsc1*

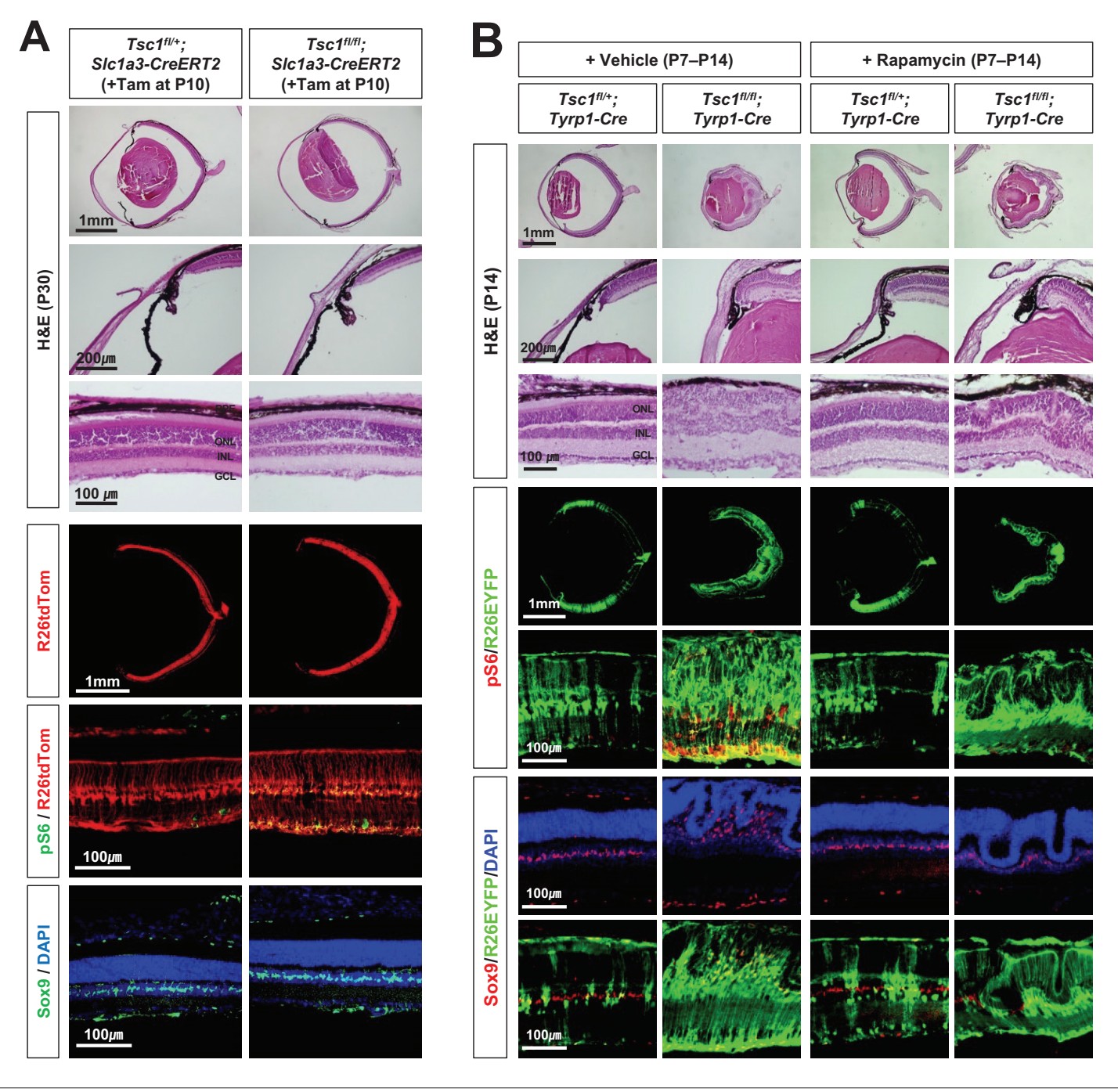

**Figure 2.** Inhibition of mTORC1 cannot suppress MG degeneration in the mature mouse retina. (**A**) The *Tsc1^{fl/+}*;*Slc1a3-CreERT2* and *Tsc1^{fl/fl}*;*Slc1a3-CreERT2* littermate mice were injected with tamoxifen (Tam) at P10 before the isolation of the eyes for cryosection at P30. The eye sections were stained with hematoxylin and eosin (H&E) to investigate the eye and retinal structures. Distributions of the cells expressing R26tdTom Cre reporter, mTORC1 activitaion marker pS6, and MG marker Sox9 in the eye sections were examined by immunostaining. Nuclei of the cells in the sections were visualized by DAPI staining. (**B**) *Tsc1^{fl/+}*;*Tyrp1-Cre* and *Tsc1^{fl/fl}*;*Tyrp1-Cre* littermate mice were injected with rapamycin (2 mg/kg) daily from P7 to P13 to inhibit mTORC1. Alternatively, the mice were injected with same volume of the vehicle (5 % poly-ethylene glycol and 5 % Tween 80 in PBS). Retinal structures of the injected mice were investigated at P14 by H&E staining of their eye sections. Distribution of Cre-affected cells and mTORC1 activation of the cells were examined by co-immunostaining of R26EYFP and pS6. Distributions of MG in the mouse retinas were examined by immunostaining of Sox9.

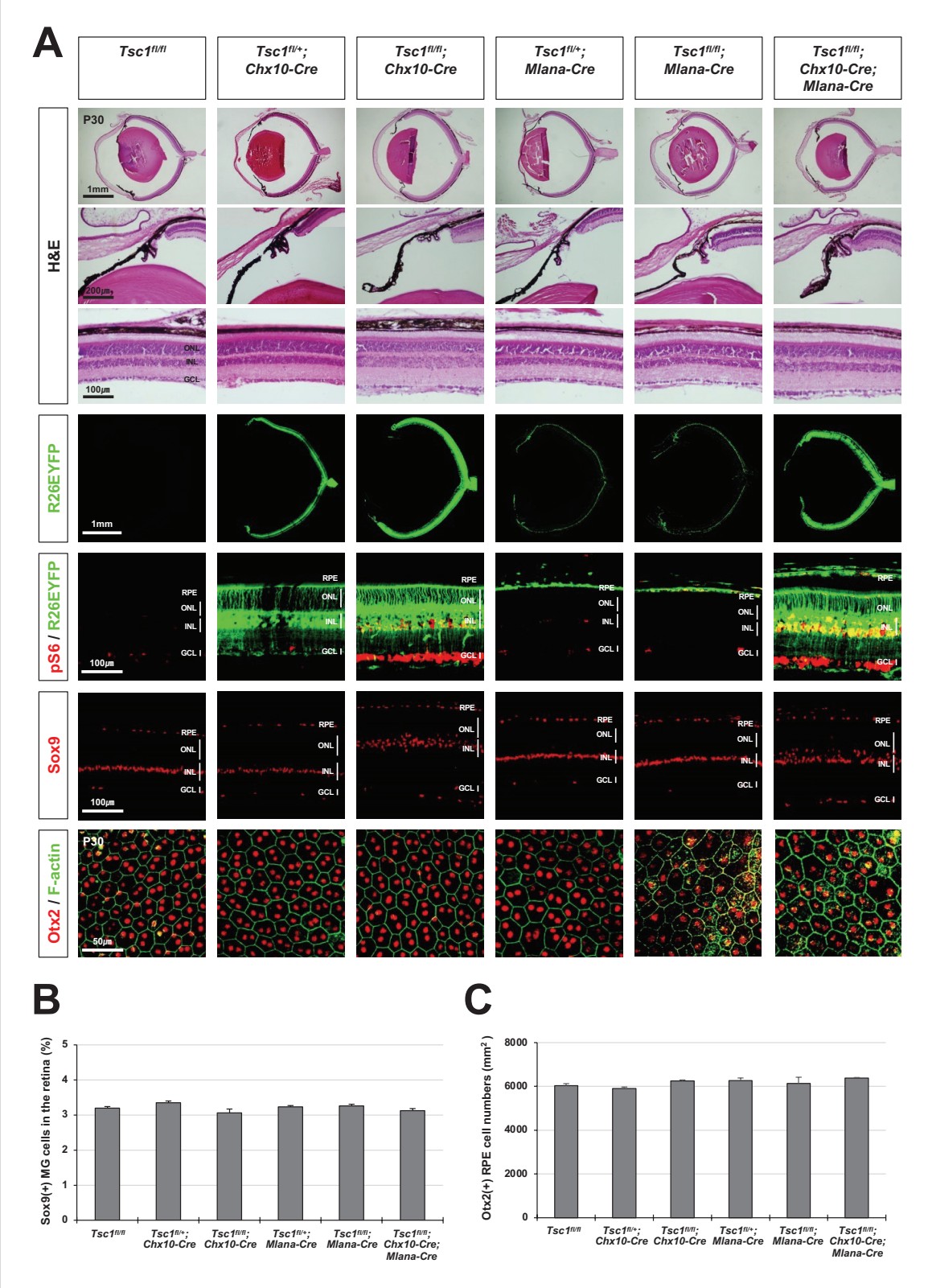

**Figure 3.** Normal eye and retinal structures in the mice lacking *tuberous sclerosis complex 1* (*Tsc1)* in majority retina and retinal pigment epithelium (RPE) populations. (**A**) Retinal structures of P30 mice deleted of *Tsc1* in majority retinal progenitor cells (RPCs) by *Chx10-Cre* or in the RPE by *Mlana-Cre* were investigated by hematoxylin and eosin (H&E) staining of the eye sections. Distribution of Cre-affected cells and mTORC1 activitation of the cells were examined by co-immunostaining of R26EYFP and pS6. Distributions of MG in the mouse retinas were examined by immunostaining of Sox9. RPE in

*Figure 3 continued on next page*

*Figure 3 continued*

whole-mount eye cups was visualized by immunostaining of Otx2, which locates in the RPE nuclei, and F-actin, which marks RPE cell boundary. Numbers of Sox9-positive MG in the retinal sections (**B**) and Otx2-postive RPE in the whole-mount eye cups (**C**) were counted and shown in the graphs. Error bars denote SD and numbers of samples are 6 from four independent litters.

was deleted in RPC and RPE together (*Figure 3A* [rightmost column] and 3B). The numbers of RPE were also unchanged in the eyes of the double Cre-expressing mice (*Figure 3A and C*).

Given the absence of Cre activity in the CM area of *Tsc1fl/fl;Chx10-Cre;Mlana-Cre* mice (see R26EYFP Cre reporter signals in *Figure 3A*), we then questioned whether the MG degeneration and rosette formation could be caused by *Tsc1* deletion in the CM population. To explore this, we deleted *Tsc1* from RPCs in the peripheral retina and the inner CM cell population using *Pax6-cre,GFP* (*Pax6-aCre*) (*Marquardt et al., 2001*). However, the eyes and retinas appeared normal in *Tsc1fl/fl;Pax6-aCre* mice, and their MG cell numbers were not greatly different from those in *Tsc1fl/+;Pax6-aCre* littermate mice (*Figure 4A*, third and fourth columns from left). Given the absence of *Pax6-aCre* activity in the pigmented outer CM cells, we next deleted *Tsc1* from the entire optic neuroepithelia-derived cell population, including the retina, the inner and outer CM, and the RPE, using *Rax-cre* (*Klimova et al., 2013*), and tested whether *Tsc1* deletion in the outer CM population is necessary for the observed phenotypes. However, *Tsc1fl/fl;Rax-Cre* mice also exhibited normal retinal morphologies and MG cell numbers (*Figure 4A*, rightmost column).

## MG degeneration is correlated with the clonal expansion of *Tsc1*-deficient RPCs in the retina

The *Tsc1*-deficient RPCs can complete a cell cycle faster than wild-type RPCs (*Choi et al., 2018*), therefore the time necessary to double the population is likely shorter for a *Tsc1*-deficient RPC than for a wild-type RPC in the same retina. This, therefore, could enable the small *Tsc1*-deficient population in the peripheral retina to expand into the central retina where wild-type populations expand slowly in *Tsc1fl/fl;Tyrp1-Cre* mice (*Figure 1B*, top row). However, a *Tsc1*-deficient RPC in *Tsc1fl/fl;Rax-Cre* and *Tsc1fl/fl;Chx10-Cre* mouse retinas might not have this competitive advantage over their neighboring RPCs for clonal expansion, since the neighboring RPCs also lose *Tsc1* and expand as quickly as the *Tsc1*-deficient RPC.

To confirm the differential clonal expansion of *Tsc1*-deficient cells in each *Tsc1-cko* mouse retina, we collected R26tdTom-positive Cre-affected cells, which are potential *Tsc1*-heterozygous and *Tsc1*-deficient cells in *Tsc1fl/+;Cre* and *Tsc1fl/fl;Cre* mouse retinas, respectively, using fluorescence-activated cell sorting (FACS). Our FACS results showed that the percentages of retinal cells affected by the Cre recombinases in *Tsc1fl/+* mouse retinas were approximately 48 % for *Tyrp1-cre*, 65 % for *Pax6-aCre*, 81 % for *Chx10-Cre*, and 92 % by *Rax-Cre*. The cells were increased to about 85 % in *Tsc1fl/fl;Tyrp1-Cre* mouse retinas; 89 % in *Tsc1fl/fl;Pax6-aCre* mouse retinas; 93 % in *Tsc1fl/fl;Chx10-Cre* mouse retinas; and 95 % in *Tsc1fl/fl;Rax-Cre* mouse retinas (*Figure 4B and C*). This resulted in 1.77-, 1.36-, 1.16-, and 1.03-fold increases, respectively, of Cre-affected cells in the *Tsc1fl/fl;Cre* mouse retinas comparing with their littermate *Tsc1fl/+;Cre* mouse retinas (*Figure 4C*). These results can also be translated to overproduction of *Tsc1*-deficient cells by 77%, 35.9%, 16.3%, and 3.2 % relative to the levels expected based on the penetrance of each Cre driver. This clonal expansion had begun during the embryonic stages and continued until P14, when retinal histogenesis was completed (*Figure 4D and E*; *Figure 4—figure supplement 1*). These results suggest the clonal expansion power of a *Tsc1*-deficient RPC is inversely correlated with the population of *Tsc1*-deficient RPCs in developing mouse retina.

Further, we deleted *Tsc1* in smaller retinal population than those affected by *Tyrp1-Cre* using *Tyrp1-CreERT2*, which can be activated in the CM and RPE subpopulation by Tam. As we expected, the R26EYFP-positive CreER-affected cells were detected sparsely in P30 *Tyrp1-CreERT2* mouse retina (*Figure 1A*), when Tam was injected by E9.5 (*Figure 4—figure supplement 2A*). We could also find retinal rosettes in the *Tsc1f/f;Tyrp1-CreERT2* mice, which were injected with Tam at E9.5, but not in those injected with Tam at P0 (*Figure 4—figure supplement 2B*). The rosette areas showed the decrease of MG and the increase of apoptotic cells (*Figure 4—figure supplement 2C–E* ). These results suggest the degeneration of MG derived from *Tsc1*-deficient CM RPCs given rise from early mouse embryos, which have greater division capacity than those from later stages.

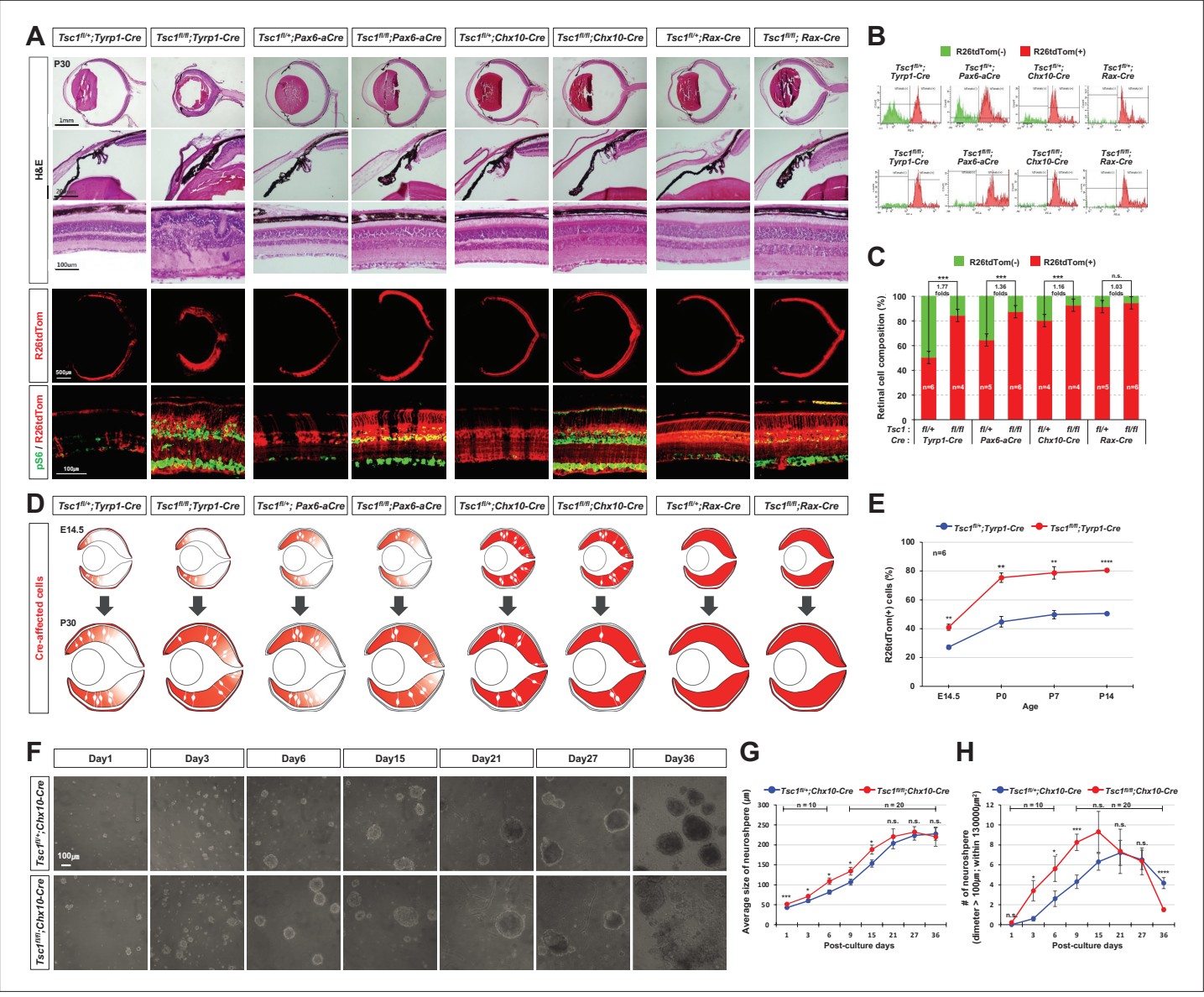

**Figure 4.** MG degeneration is caused by clonal hyperexpansion of *tuberous sclerosis complex 1* (*Tsc1*)-deficient cells in developing mouse retina. (**A**) Eye and retinal structures of P30 mice with the indicated genotypes were examined by hematoxylin and eosin (H&E) staining. Distribution of Cre-affected cells and mTORC1 activation of the cells were examined by co-immunostaining of R26tdTom and pS6. (**B**) Cre-affected *Tsc1*-heterozygote (*Tsc1*^fl/+^;*Cre*) and *Tsc1*-deficient (*Tsc1*^fl/fl^;*Cre*) cells, which emitted red fluorescence of R26tdTom Cre reporter, are isolated from Cre-unaffected wild-type cells in same retinas by FACS and the histograms are presented. (**C**) Relative composition of R26tdTom-positive and -negative cell populations in the retinas are shown in the graph. Error bars denote SD. Numbers of samples analyzed are shown in the graph columns. ***$p < 0.001$; n.s., not significant. (**D**) Distributions of the Cre-affected cells in the corresponding E14.5 and P30 mouse retinas are summarized in the drawings. (**E**) Developmental changes of R26tdTom-positive Cre-affected population in *Tsc1*^fl/+^;*Tyrp1-Cre* and *Tsc1*^fl/fl^;*Tyrp1-Cre* littermate mouse retinas were determined by FACS and shown in the graph. Error bars denote SD. Numbers of samples analyzed are shown in the graph (four independent litters). **$p < 0.01$; ****$p < 0.0001$. (**F**) *Tsc1*^+/-^ and *Tsc1*^-/-^ retinal progenitor cells (RPCs) were isolated from E13 *Tsc1*^fl/+^;*Chx10-Cre* and *Tsc1*^fl/fl^;*Chx10-Cre* and were cultured to form the neurospheres. Representative images of the neurospheres at the indicated post-culture days are provided. (**G**) Sizes of neurospheres in culture were counted at the indicated post-culture days and their average values are shown in the graph. (**H**) Average numbers of neurosphere in the indicated area are also shown in the graph. Numbers of samples analyzed are shown in the graphs (± independent culture batches). *$p < 0.05$; ***$p < 0.001$; ****$p < 0.0001$; n.s., not significant.

The online version of this article includes the following figure supplement(s) for figure 4:

**Figure supplement 1.** Distributions of *tuberous sclerosis complex 1* (*Tsc1*)-deficient cells in E14.5 *Tsc1-conditional knock-out (cko)* mouse retinas.

**Figure supplement 2.** MG degeneration and retinal rosette formation upon the deletion of *tuberous sclerosis complex 1* (*Tsc1*) in ciliary margin (CM)-derived minor retinal progenitor cell (RPC) clones by tyrosinase-related protein 1-Cre (Tyrp1-Cre)ERT2.

*Figure 4 continued on next page*

*Figure 4 continued*

**Figure supplement 3.** Autonomous effects of *tuberous sclerosis complex 1* (*Tsc1*) deletion on mitotic aging of mouse retinal progenitor cells (RPCs) in neurosphere culture.

**Figure supplement 4.** Hypothetical model depicts the clonal hyperexpansion of *tuberous sclerosis complex 1* (*Tsc1*)-deficient.

## Earlier arrival of *Tsc1*-deficent mouse RPCs at the division limit

Given the positive relationship between the clonal expansion power of *Tsc1*-deficient cells and MG degeneration in the *Tsc1-cko* mouse retinas, we hypothesized that the *Tsc1*-deficient RPCs in *Tsc1*<sup>fl/fl</sup>;*Tyrp1-Cre* mouse retinas were likely to be too old, in mitotic terms, to preserve the survival program intact in their descendent MG. In the other words, the *Tsc1*-deficient RPCs were close to or had exceeded their mitotic division limit when they produced MG in *Tsc1*<sup>fl/fl</sup>;*Tyrp1-Cre* mouse retinas, whereas the RPCs in the other *Tsc1-cko* mice had not yet reached that limit.

To test this hypothesis, we compared the division capacities of *Tsc1*-heterozygote and *Tsc1*-deficient mouse RPCs in vitro. The RPCs were isolated from E13 *Tsc1*<sup>fl/+</sup>;*Chx10-Cre* and *Tsc1*<sup>fl/fl</sup>;*Chx10-Cre* littermate mouse embryo retinas, respectively, and cultured to form neurospheres. We found that *Tsc1*-deficient RPCs divided faster to form larger neurospheres than those derived from *Tsc1*-heterozygote RPCs at one post-culture week (*Figure 4F*). However, the *Tsc1*-deficient neurospheres stopped expanding and the cells comprising the neurospheres started to degenerate after 3 post-culture weeks, while the *Tsc1*-heterozygote neurospheres expanded slowly and maintained in relative intact forms by 5 post-culture weeks (*Figure 4F and G*). Consequently, the numbers of *Tsc1*-deficient neurospheres became less than those of *Tsc1*-heterozygote neurospheres after 4 post-culture weeks (*Figure 4F and H*). The fast expansion and precocious degeneration of *Tsc1*-deficient neurospheres were not affected by the presence of wild-type neurospheres in the neighbor, suggesting that over-proliferation is an intrinsic property of *Tsc1*-deficient RPCs (*Figure 4—figure supplement 3A – D*). These results suggest that the RPC has a limited division capacity that can be reachable earlier when the cells divide faster, as seen in the *Tsc1*-deficient RPCs (*Figure 4—figure supplement 4*).

## Accelerated mitotic aging of *Tsc1*-deficient mouse RPCs

To further validate the idea that the *Tsc1*-deficient RPCs in *Tsc1*<sup>fl/fl</sup>;*Tyrp1-Cre* mice overproliferated to produce MG after exceeding their division limits, we explored the relative division numbers of retinal cells comprising various *Tsc1-cko* mouse retinas. Telomeric DNA sequences are shortened after every cell division unless they are recovered by telomerase (*Calado and Dumitriu, 2013*; *Shay, 2016*). Given the absence of *telomerase reverse transcriptase* (*Tert*) and *telomerase RNA component* (*Terc*) mRNA expression in embryonic and postnatal mouse retinas (data not shown), we expected that the telomeric sequences would be shortened constantly in mouse RPCs after each division. In support of this, we found that the average telomere length in mouse retina cells decreased with age (*Figure 5—figure supplement 1*). We further found that the telomere shortening was faster in *Tsc1*<sup>fl/fl</sup>;*Tyrp1-Cre* mouse retinas compared to *Tsc1*<sup>fl/+</sup>;*Tyrp1-Cre* littermate mouse retinas (*Figure 5A*; *Figure 5—figure supplement 1*). However, there was no significant difference in telomere length between other *Tsc1*<sup>fl/+</sup>;*Cre* and *Tsc1*<sup>fl/fl</sup>;*Cre* littermate mouse retinas (*Figure 5A*). These results suggest the ability of *Tsc1*-deficient RPCs to overproliferate is inversely related with their frequencies in the retina.

To examine whether telomere shortening was an autonomous event of *Tsc1*-deficient cells, we next compared the telomere lengths of FACS-isolated wild-type and *Tsc1*-deficent cells in the same *Tsc1*<sup>fl/fl</sup>;*Tyrp1-Cre* mouse retinas. We found that the telomeres in R26tdTom-positive *Tsc1*-deficient cells of P14 *Tsc1*<sup>fl/fl</sup>;*Tyrp1-Cre* mouse retinas were significantly shorter than those in R26tdTom-negative neighboring wild-type cells (*Figure 5B*). The accelerated telomere shortening was also observed in the neurospheres derived from *Tsc1*-deficent RPCs, which were cultured homogenously or together with wild-type neurospheres (*Figure 4—figure supplement 3F*). These results suggest that *Tsc1* deletion had autonomous effects on telomere shortening.

## Senescence-associated cell death of MG in hyperexpanded *Tsc1*-deficient retinal clones

Extensive telomere shortening can cause mitotic catastrophe and subsequent cell death (*Calado and Dumitriu, 2013*; *Shay, 2016*), suggesting that the degeneration of MG in *Tsc1*<sup>fl/fl</sup>;*Tyrp1-Cre* mouse

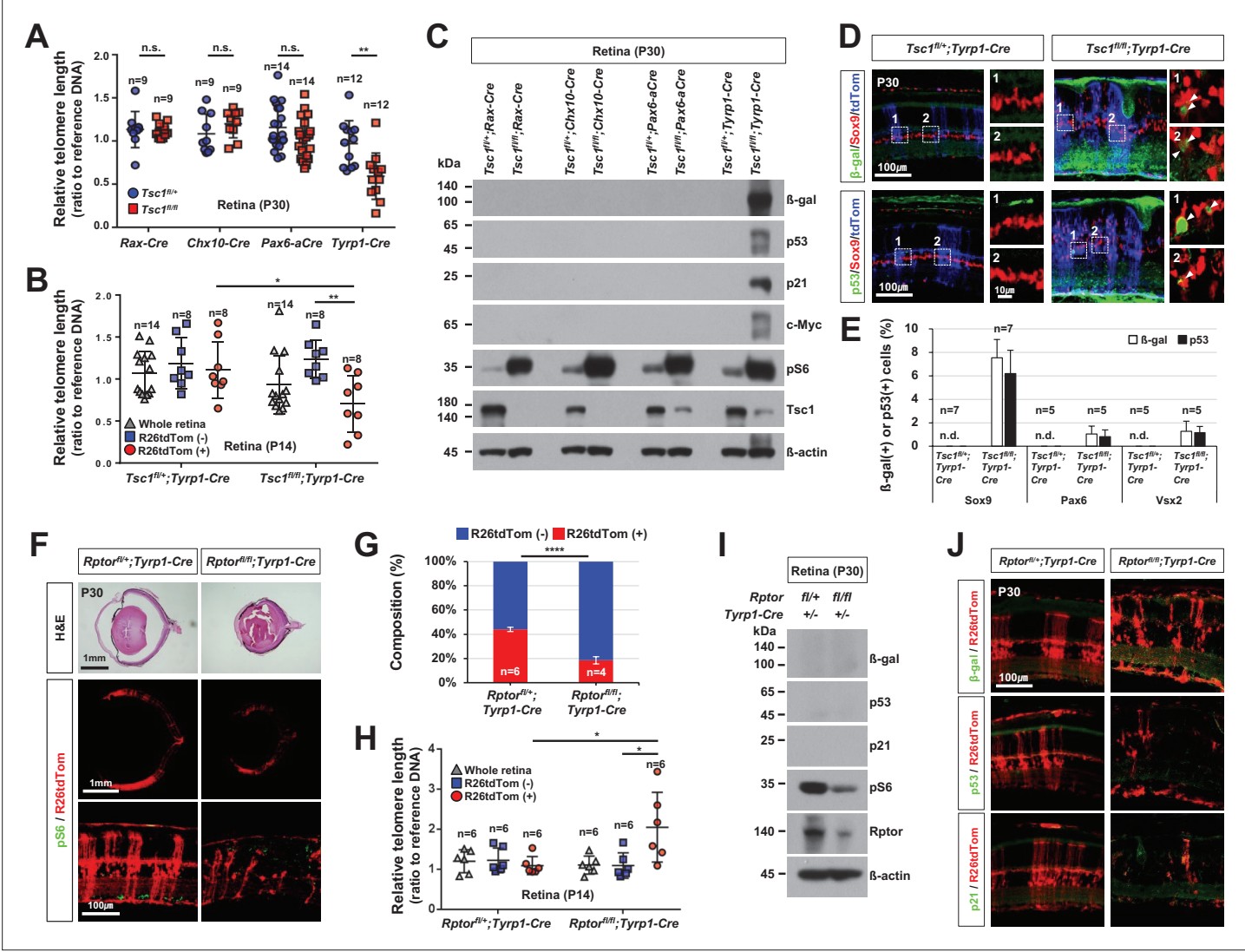

**Figure 5.** Mitotic aging and senescence of *tuberous sclerosis complex 1* (*Tsc1*)-deficient retinal cells. (**A**) Telomere lengths of the retinal cells, which were isolated from P30 *Tsc1^{fl/+}* and *Tsc1^{fl/fl}* mice with the indicated Cre drivers, were compared with that of control sample, and relative values are shown in the graph. Error bars denote SD. Numbers of samples analyzed are shown in the graph. All samples were obtained from independent litters. **p < 0.01; n.s., not significant. (**B**) Telomere lengths of the FACS-isolated retinal cells from P14 *Tsc1^{fl/+};Tyrp1-Cre* and *Tsc1^{fl/fl};Tyrp1-Cre* littermate mice were compared with that of control sample, and their relative values are shown in the graph. *p < 0.05; ***p < 0.001. (**C**) Senescence markers ($\beta$-gal, p53, p21, and c-Myc) expressed in the corresponding P30 *Tsc1^{fl/+};Cre* and *Tsc1^{fl/fl};Cre* mouse retinas were detected by Western blot (WB). Relative amounts of the proteins used in each sample were determined by WB detection of $\beta$-actin. (**D**) Expression of senescence markers ($\beta$-gal and p53) in the MG was determined by co-immunostaining with an MG marker, Sox9. The cells experienced Cre-mediated recombination were also visualized by R26tdTom reporter. Images in the right columns are the magnified versions of the boxed areas indicated by corresponding numbers in the low magnification images. (**E**) Sox9-positive MG, Pax6-positive amacrine cell (AC), and Vsx2-positive bipolar cell (BC) populations that express the senescence markers were determined and shown in the graph (Pax6 and Vsx2 staining images are provided in *Figure 5—figure supplement 4*). Number of samples analyzed are shown in the graph (five independent litters). n.d., not detected. (**F**) Eye and retinal structures of P30 *Rptor^{fl/+};Tyrp1-Cre* and *Rptor^{fl/fl};Tyrp1-Cre* mice were investigated by hematoxylin and eosin (H&E) staining of the eye sections. Distribution of the Cre-affected cells and mTORC1 activation of the cells were examined by the immunostaining of R26tdTom and pS6, respectively. (**G**) R26tdTom-positive Cre-affected cell population in the mouse retinas were quantified by FACS and shown in the graph. ****p < 0.0001. (**H**) Telomere lengths of unsorted (whole retina) and the FACS-isolated retinal cells from P14 *Rptor^{fl/+};Tyrp1-Cre* and *Rptor^{fl/fl};Tyrp1-Cre* mice were compared with that of control sample and their relative values are shown in the graph. Number of samples analyzed are shown in the graph. *p < 0.05. (**I**) Relative levels of senescence markers and mTORC1 pathway components of the mouse retinas were analyzed by WB. (**J**) Distribution of the cells expressing senescence markers and R26tdTom Cre reporter in the retinas was also examined by immunostaining.

The online version of this article includes the following figure supplement(s) for figure 5:

**Figure supplement 1.** Telomere shortening in developing mouse retina.

*Figure 5 continued on next page*

*Figure 5 continued*

**Figure supplement 2.** Telomerase overexpression could not rescue the phenotypes in *Tsc1^fl/fl^;Tyrp1-Cre* mouse retinas.

**Figure supplement 3.** Expression of senescence markers in the postnatal *tuberous sclerosis complex 1* (*Tsc1*)- and *Tsc2-conditional knock-out (cko)* mouse retinas.

**Figure supplement 4.** Senescence is not evident in amacrine cells (ACs) and bipolar cells (BCs) of *Tsc1^fl/fl^;Tyrp1-Cre* mouse retina.

retina could be caused by telomere shortening beyond a critical length. To test this possibility, we bred *Tsc1^fl/fl^;Tyrp1-Cre* mice with *Cre-d-Tert* (*dTert*) transgenic mice, which express mouse *Tert* cDNA in Cre-affected cells (*Hidema et al., 2016*), to recover telomeres in *Tsc1*-deficient retinal cells (*Figure 5—figure supplement 2C*). However, MG degeneration and retinal rosettes were still observed in *Tsc1^fl/fl^;Tyrp1-cre;dTert* mouse retinas (*Figure 5—figure supplement 2A and B*). This suggests that telomere shortening is unlikely a direct cause of MG degeneration in *Tsc1^fl/fl^;Tyrp1-Cre* mice.

The more cells divide the more damage accumulates, resulting in senescence-associated cell death (*Shay, 2016*). We found that markers of senescence, including β-galactosidase (β-gal), transformation-related protein 53 (Trp53, p53), p21 cyclin-dependent kinase inhibitor 1 A (p21^Cip1^, p21), and c-Myc (*Kuilman et al., 2010*), were detected in P30 *Tsc1^fl/fl^;Tyrp1-Cre* mouse retinas but not in *Tsc1^fl/+^;Tyrp1-Cre* littermate mouse retinas (*Figure 5C*; *Figure 5—figure supplement 3A*). Furthermore, β-gal and p53 were enriched more in R26tdTom-positive *Tsc1*-deficient MG (*Figure 5*; *Figure 5—figure supplement 4*), which degenerated to form the retinal rosettes (*Figure 1C,D*), than in R26tdTomato-positive *Tsc1*-deficient AC (Pax6-positive) and BC (Vsx2-positive) populations, which remained intact in the retinas (*Figure 1D*; *Figure 1—figure supplement 2A*). The senescence markers were also increased in *Tsc2^fl/fl^;Tyrp1-Cre* mouse retinas (*Figure 5—figure supplement 3B*), suggesting that the retinal senescence occurred in an mTORC1-dependent manner. In contrast, the senescence markers were not detectable in *Tsc1^fl/fl^;Rax-Cre*, *Tsc1^fl/fl^;Chx10-Cre*, and *Tsc1^fl/fl^;Pax6-aCre* mouse retinas (*Figure 5C*), suggesting a positive relationship between RPC clonal expansion power and senescence.

In contrast to the accelerated cell cycle progression of *Tsc1*-deficient RPCs, mouse RPCs lacking *Rptor* (*regulatory-associated protein of mTOR*), a key component of mTORC1 (*Kim et al., 2002*), were found to exit the cell cycle precociously (*Choi et al., 2018*). Consequently, the *Rptor*-deficient RPCs failed to expand, resulting in the compression of R26tdTom-positive *Rptor*-deficient clones in *Rptor^fl/fl^;Tyrp1-Cre* mouse retinas compared to those in *Rptor^fl/+^;Tyrp1-Cre* littermate mouse retinas (*Figure 5F and G*). These results suggest that retinal cells derived from the *Rptor*-deficient RPCs might be mitotically younger than neighboring wild-type cells in the *Rptor^fl/fl^;Tyrp1-Cre* mouse retinas.

Thus, we compared the relative telomere lengths of FACS-isolated *Rptor*-deficient cells and wild-type cells in the same retinas. We found that R26tdTom-positive *Rptor*-deficient cells of P14 *Rptor^fl/fl^;Tyrp1-Cre* mouse retinas had longer telomeres than R26tdTom-negative wild-type neighbors and R26tdTom-positive *Rptor*-heterozygote cells of *Rptor^fl/+^;Tyrp1-Cre* littermate mouse retinas (*Figure 5H*). However, we did not observe a significant difference in the telomere lengths of R26tdTom-negative wild-type cells in *Rptor^fl/fl^;Tyrp1-Cre* and *Rptor^fl/+^;Tyrp1-Cre* littermate mouse retinas, suggesting that there was no compensatory overproliferation of wild-type RPCs after the depletion of *Rptor*-deficient RPCs in *Rptor^fl/fl^;Tyrp1-Cre* mouse retinas. There was also no increase in the senescence markers in *Rptor^fl/fl^;Tyrp1-Cre* mouse retinas, either (*Figure 5I and J*). Together, our results suggest that senescence occurs in retinas where RPCs have exceeded their division limits, as seen in *Tsc1^fl/fl^;Tyrp1-Cre* mice, but not in retinas where RPCs have not reached the limits, as seen in the other *Tsc1-cko* and *Rptor^fl/fl^;Tyrp1-Cre* mice.

## *Hif1a* is necessary for the hyperexpansion of *Tsc1*-deficient retinal clones

We next sought for the strategies to suppress the RPC overproliferation that led to the clonal hyperexpansion of *Tsc1*-deficient cells and consequent degeneration of MG in *Tsc1^fl/fl^;Tyrp1-Cre* mouse retinas. We previously showed that mTORC1 induces immunoproteasomes to increase the turnover of cyclins and thereby accelerate RPC cell cycle progression (*Choi et al., 2018*). We thus deleted the three catalytic subunit genes, *proteasome subunit beta 8, 9*, and *10* (*Psmb8,9,10*), to eliminate the contribution of immunoproteasomes to the mTORC1-induced mitotic acceleration of *Tsc1*-deficient mouse RPCs (*Figure 6—figure supplements 1 and 2B*). We found that retinal rosettes were absent

from approximately a quarter of in *Psmb8⁻/⁻,9⁻/⁻,10⁻/⁻ (Psmb-tko);Tsc1^fl/fl;Tyrp1-Cre* mouse retinas, and R26tdTom-positive *Tsc1*-deficient retinal clone size was decreased in comparison with that of *Tsc1^fl/fl;Tyrp1-Cre* mouse retinas (*Figure 6—figure supplement 2A* [rightmost column] and 2 C). However, retinal rosettes and *Tsc1*-deficient retinal clonal expansion were still observed in about three-quarters of *Psmb-tko;Tsc1^fl/fl;Tyrp1-Cre* mouse retinas (*Figure 6—figure supplement 2A* [second rightmost column] and 2 C). These results suggest that there are additional regulator(s) of mTORC1-induced RPC clonal expansion, in addition to the immunoproteasomes.

In parallel to promoting protein degradation via the immunoproteasomes, mTORC1 also enhances the synthesis of many cellular proteins by activating S6 via S6K1 and by directly inactivating eukaryotic translation initiation factor 4E-binding protein 1 (*Ben-Sahra and Manning, 2017*; *Saxton and Sabatini, 2017*). Hif1a is among the proteins subjected to mTORC1-induced translational upregulation (*Bernardi et al., 2006*); it, in turn, induces the expression of various target genes to support mTORC1-induced cell proliferation, growth, and survival (*Keith et al., 2011*). We also found that the

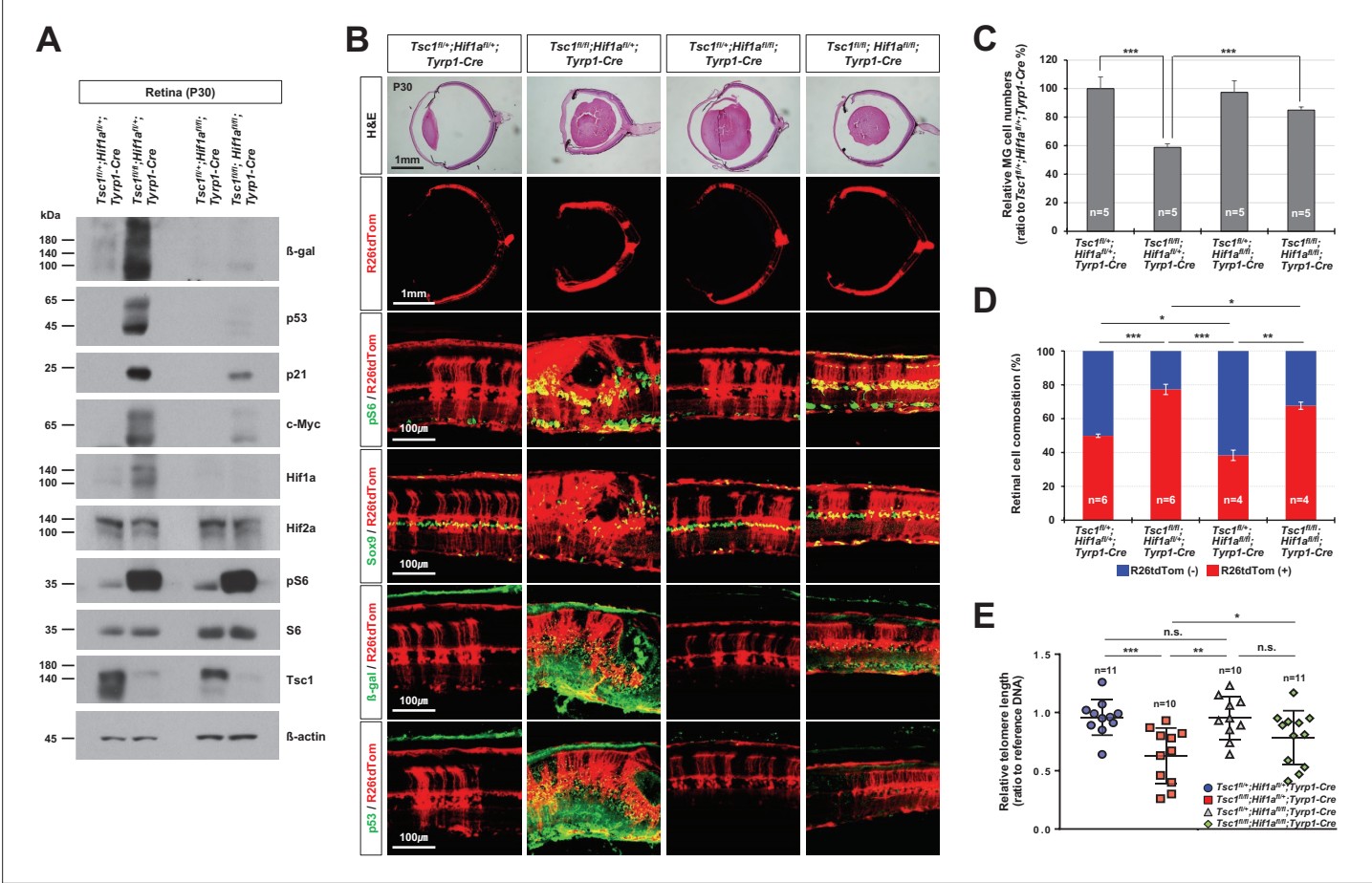

**Figure 6.** Rescue of *Tsc1^fl/fl;Tyrp1-Cre* mouse retinal phenotypes by concomitant deletion of *hypoxia-induced factor 1-alpha* (*Hif1a*). (**A**) Levels of senescence markers in P30 mouse retinas with indicated genotypes were analyzed by Western blot (WB). Relative amounts of proteins used in each sample were determined by WB detection of β-actin. (**B**) Distributions of mTORC1-active cells, which are positive to pS6, MG, which are positive to Sox9, and senescent cells, which are positive to β-gal and p53, were examined by immunostaining of the eye sections. Expression of those markers in the Cre-affected cells were determined by comparing the expression of the Cre reporter R26tdTom. (**C**) Numbers of Sox9-positive MG in the mouse retinas were counted and the relative numbers are shown in the graph. (**D**) R26tdTom-negative wild-type cells and R26tdTom-positive Cre-affected cells in the retinas were determined byFACS and shown in the graph. (**E**) The lengths of telomeres of the cells isolated from P30 mouse retinas with the indicated genotypes were compared with that of control sample, and their relative values are shown in the graph. Error bars denote SD. Numbers of samples analyzed are shown in the graphs. n.s., not significant; *p < 0.05; **p < 0.01; ***p < 0.001.

The online version of this article includes the following figure supplement(s) for figure 6:

**Figure supplement 1.** CRISPR/Cas9 gene targeting of mouse *proteasome subunit beta 8*, *9*, and *10* (*Psmb8,9,10*) genes.

**Figure supplement 2.** Immunoproteasome deficiency rescues the retinal phenotypes of *Tsc1^fl/fl;Tyrp1-Cre* mice incompletely.

level of Hif1a, but not the level of its homolog, Hif2a, was significantly increased in *Tsc1*<sup>fl/fl</sup>;*Tyrp1-Cre* mouse retinas (***Figure 6A***). To investigate whether mTORC1 mediated Hif1a to generate the observed phenotypes, we deleted *Hif1a* concomitantly with *Tsc1* in *Tsc1*<sup>fl/fl</sup>;*Hif1a*<sup>fl/fl</sup>;*Tyrp1-Cre* mice. We found that 96 % of *Tsc1*<sup>fl/fl</sup>;*Hif1a*<sup>fl/fl</sup>;*Tyrp1-Cre* mouse retinas did not have rosette structures and MG cell loss (***Figure 6B*** [third row] and C). R26tdTom-positive Cre-affected cell population was also decreased significantly in *Tsc1*<sup>fl/fl</sup>;*Hif1a*<sup>fl/fl</sup>;*Tyrp1-Cre* mouse retinas compared to *Tsc1*<sup>fl/fl</sup>;*Tyrp1-Cre* littermate mouse retinas (***Figure 6B*** [second row] and D), suggesting that the hyperexpansion of *Tsc1*-deficient cell clones was suppressed in these retinas. Furthermore, the telomere lengths of *Tsc1*<sup>fl/fl</sup>;*Hif1a*<sup>fl/fl</sup>;*Tyrp1-Cre* mouse retinal cells were increased relative to those of *Tsc1*<sup>fl/fl</sup>;*Tyrp1-Cre* littermate mouse retinal cells (***Figure 6E***). The levels of the senescence markers induced in degenerating *Tsc1*<sup>fl/fl</sup>;*Tyrp1-Cre* mouse retinas were also decreased significantly in *Tsc1*<sup>fl/fl</sup>;*Hif1a*<sup>fl/fl</sup>;*Tyrp1-Cre* mouse retinas (***Figure 6A and B*** [two bottom rows]). These results suggest that *Hif1a* is necessary for the clonal hyperexpansion of *Tsc1*-deficient cells and the subsequent mitotic aging and senescence-associated degeneration of MG in *Tsc1*<sup>fl/fl</sup>;*Tyrp1-Cre* mouse retina.

## *Hif1a* supports the accelerated cell cycle progression of *Tsc1*-deficient RPCs

We next investigated whether *Hif1a* supports the overproliferation of *Tsc1*-deficient RPCs and thus the hyperexpansion of *Tsc1*-deficient retinal clones. We measured the numbers of proliferating cells that incorporated a thymidine analog 5-ethynyl-2′-deoxyuridine (EdU) during S-phase of the cell cycle and then progressed to G2/M-phases by accumulating phosphorylated histone H3 (pH3) or incorporated two thymidine analogs, 5-chloro-2'-deoxyuridine (CldU) and 5-iodo-2'-deoxyuridine (IdU), serially in their first and second S-phases (***Figure 7A***). We found that the numbers of EdU;pH3-positive and CldU;IdU-positive RPCs were significantly elevated in P0 *Tsc1*<sup>fl/fl</sup>;*Tyrp1-Cre* mouse retinas in comparison to those in *Tsc1*<sup>fl/+</sup>;*Tyrp1-Cre* littermate mouse retinas (***Figure 7A*** [third and seventh rows] and C). The numbers of EdU;Otx2-positive cells, which exited the cell cycle for differentiation to PRs in 12 hr, were also increased in the *Tsc1*<sup>fl/fl</sup>;*Tyrp1-Cre* mouse retinas (***Figure 7A*** [fifth row] and C). Consequently, the sizes of R26tdTom-positive *Tsc1*-deficient cells in P0 *Tsc1*<sup>fl/fl</sup>;*Tyrp1-Cre* mouse retinas were significantly increased compared to those of R26tdTom-positive *Tsc1*-heterozygous retinal cells in *Tsc1*<sup>fl/+</sup>;*Tyrp1-Cre* littermates (***Figure 7A*** [top row] and B). However, the numbers of proliferating cells and the sizes of R26tdTom-positive *Tsc1*-deficient clones were decreased significantly in P0 *Tsc1*<sup>fl/fl</sup>;*Hif1a*<sup>fl/fl</sup>;*Tyrp1-Cre* littermate mouse retinas (***Figure 7A*** [right most columns] - C). These results therefore suggest that *Hif1a* is necessary for the accelerated cell cycle progression of *Tsc1*-deficient mouse RPCs and consequent expansion of *Tsc1*-deficient retinal clones.

## Hypoxia and induction of *Hif1a* in *Tsc1*<sup>fl/fl</sup>;*Tyrp1-Cre* mouse retinas

We found that the level of Hif1a protein, but not that of *Hif1a* mRNA, was increased significantly in P0 *Tsc1*<sup>fl/fl</sup>;*Tyrp1-Cre* mouse retinas (***Figure 7D and E***). Hif1a is known to be accumulated under hypoxia by being resistant from proteasomal degradation (***Ivan et al., 2001***; ***Jaakkola et al., 2001***). Therefore, Hif1a elevation could result from mTORC1-induced translational upregulation and/or hypoxia. We, thus, examined whether a hypoxic condition was established in the mouse retinas using the hypoxy-probe (***Varghese et al., 1976***), and found that P0 *Tsc1*<sup>fl/fl</sup>;*Tyrp1-Cre* mouse retinas were more hypoxic than *Tsc1*<sup>fl/+</sup>;*Tyrp1-Cre* littermate retinas (***Figure 7E and G*** [bottom row images]). Interestingly, although hypoxia can compromise mitochondrial ATP synthesis (***Vander Heiden et al., 2009***), the level of ATP was significantly elevated in P0 *Tsc1*<sup>fl/fl</sup>;*Tyrp1-Cre* mouse retinas compared to that in *Tsc1*<sup>fl/+</sup>;*Tyrp1-Cre* littermate control retinas, and it returned to the control level in P0 *Hif1a*<sup>fl/fl</sup>;*Tsc1*<sup>fl/fl</sup>;*Tyrp1-Cre* mouse retinas (***Figure 7—figure supplement 1A***). These results suggest that *Tsc1*-deficient RPCs mediate Hif1a to supply the ATP, which is necessary for their accelerated cell cycle progression and cell growth under the hypoxic condition of *Tsc1*<sup>fl/fl</sup>;*Tyrp1-Cre* mouse retinas.

## Hif1a-induced glycolytic gene expression is necessary for the proliferation of Tsc1-deficient RPCs

It is known that glycolysis plays important roles in cellular ATP synthesis under hypoxia (***Cerychova and Pavlinkova, 2018***). Consistent with this, the level of lactate, which accumulates when glycolysis occurs in a condition of inefficient mitochondrial respiration (***Cerychova and Pavlinkova, 2018***),

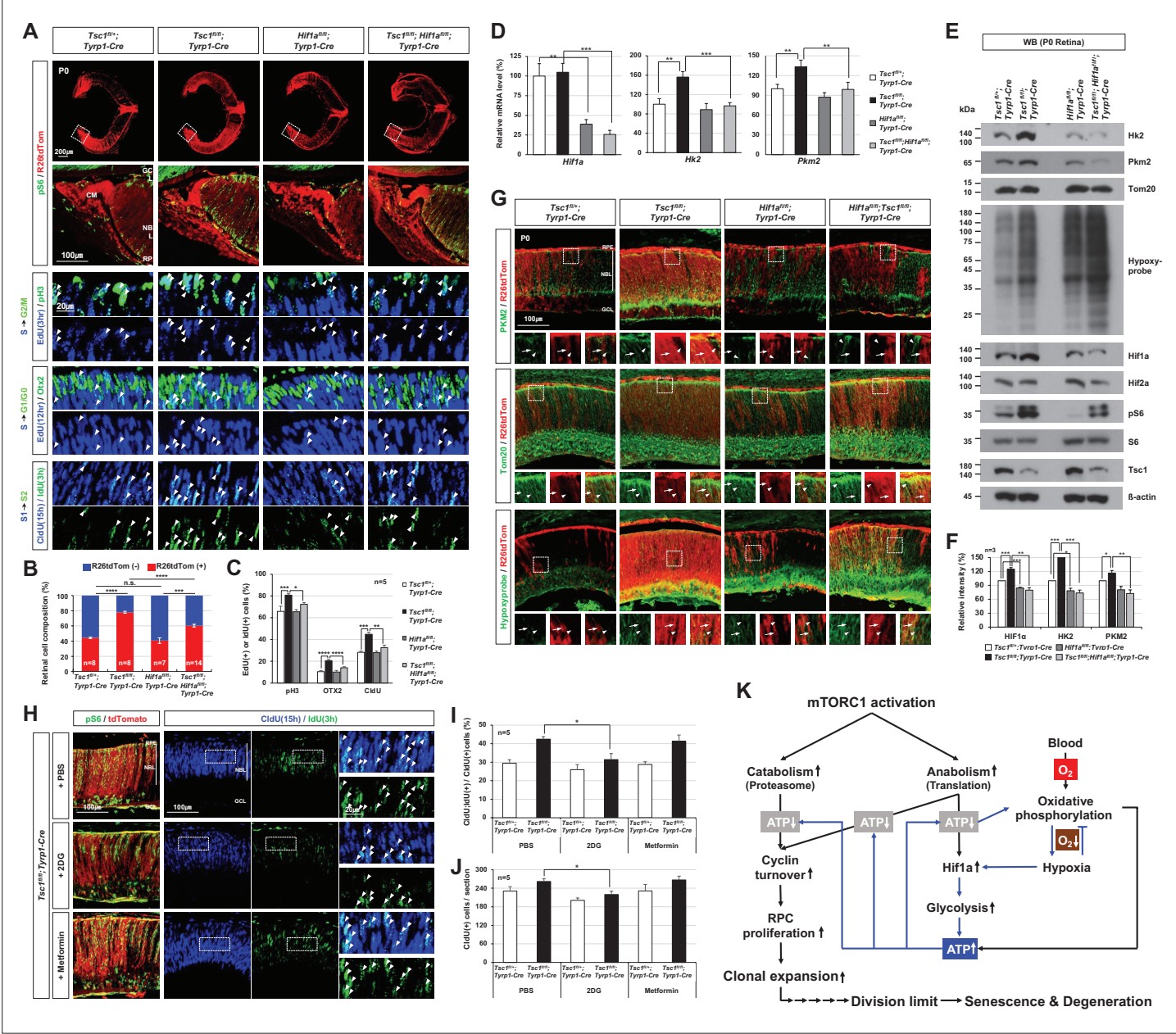

**Figure 7.** Hif1a supports mTORC1-induced retinal progenitor cell (RPC) proliferation through the expression of glycolytic enzymes. (**A**) The Cre-affected cells in P0 mouse retinas with the indicated genotypes were visualized by R26tdTom Cre reporter. Distribution of mTORC1-active cells in the boxed areas was also examined by immunostaining of pS6 and shown in the second row. Proliferation and cell cycle progression of RPCs in the mouse retinas were also examined by 5-ethynyl-2'-deoxyuridine (EdU)/5-chloro-2'-deoxyuridine (CldU) labeling and chasing experiments as described in Materials and methods. (**B**) R26tdTom-positive Cre-affected cell population in the mouse retinas were quantified by FACS and shown in the graph. Error bars, SD. Numbers of samples analyzed are shown in the graph (four independent litters). (**C**) RPCs that had progressed from S-phase to G2/M-phase for 3 hr (EdU;pH3-positive); PRs that had been born for 12 hr (EdU;Otx2-positive); and RPCs that had reentered cell cycle for 15 hr (CldU;IdU-positive) in the mouse retinas were counted and shown in the graph. Error bars denote SD (n = 5, five independent litters). *p < 0.05; **p < 0.01; ***p < 0.001; ****p < 0.0001. (**D**) Relative mRNA levels of *Hif1a*, hexokinase 2 (*Hk2*), and pyruvate kinase M2 (*Pkm2*) in P0 mouse retinas with the indicated genotypes were examined by real-time quantitative polymerase chain reaction (RT-qPCR). The y-axis values are relative $2^{-\Delta Ct}$ values against those of *Tsc1^{fl/+};Tyrp1-Cre* mouse retinas. Error bars, SD (n = 4, three independent litters). (**E**) Expressions of the indicated proteins in the mouse retinas were examined by Western blot (WB). Relative amounts of the proteins used in each sample were determined by WB detection of $\beta$-actin. (**F**) Relative WB band intensities of the proteins are determined by the ImageJ software and shown in the graph. Error bars, SD (n = 3, three independent litters). (**G**) Distributions of Pkm2, Tom20, and hypoxyprobe-labeled proteins in the mouse retinas were examined by immunostaining. (**H**) P0 *Tsc1^{fl/fl};Tyrp1-Cre* littermate mice were injected with a chemical inhibitor of glycolysis (2DG) or mitochondrial oxidative phosphorylation (Metformin), and the eye sections of the mice were

*Figure 7 continued on next page*

Figure 7 continued

obtained at 12 hr post-injection for the immunodetection of RPCs that had proliferated (CldU-positive) and/or reentered second cell cycle (CldU;IdU-positive) for 15 hr. Relative numbers of those cells in the mouse retinas were shown in the graphs in (I) and (J), respectively. Error bars, SD (n = 5, four independent litters). (K) Schematic diagram depicts the mTORC1-Hif1a-glycolysis cascade in developmental RPC clonal hyperexpansion, which leads to senescence-associated degeneration of MG in the mature retina.

The online version of this article includes the following figure supplement(s) for figure 7:

**Figure supplement 1.** Enhanced glycolytic ATP synthesis in *Tsc1fl/fl;Tyrp1-Cre* mouse retinas.

was significantly elevated in P0 *Tsc1fl/fl;Tyrp1-Cre* mouse retinas compared to *Tsc1fl/+;Tyrp1-Cre* littermate mouse retinas, but not in *Hif1afl/fl;Tsc1fl/fl;Tyrp1-Cre* littermate mouse retinas (*Figure 7—figure supplement 1B*). This suggests that Hif1a is necessary for the enhanced glycolysis observed in *Tsc1fl/fl;Tyrp1-Cre* mouse retinas.

We further examined whether Hif1a contributes to glycolytic ATP synthesis in *Tsc1*-deficient cells by inducing the expression of the glycolytic enzymes, such as *hexokinase 2* (*Hk2*) and *pyruvate kinase M2* (*Pkm2*), which are transcriptional targets of Hif1a (*Düvel et al., 2010*). We found that the mRNA and protein levels of those glycolytic enzymes were significantly elevated in P0 *Tsc1fl/fl;Tyrp1-Cre* mouse retinas compared to *Tsc1fl/+;Tyrp1-Cre* littermate retinas (*Figure 7D-F*). The enzyme levels in P0 *Tsc1fl/fl;Hif1afl/fl;Tyrp1-Cre* mouse retinas were decreased and became similar to those in P0 *Tsc1fl/+;Tyrp1-Cre* mouse retinas. Our findings suggest that Hif1a plays a critical role in elevating these glycolytic enzymes in *Tsc1fl/fl;Tyrp1-Cre* mouse retinas.

Next, to delineate the causal relationship between glycolysis and the hyperproliferation of *Tsc1*-deficient RPCs, we injected 2-deoxy-D-glucose (2DG), a chemical inhibitor of glycolysis (*Wick et al., 1957*), or metformin, a chemical inhibitor of the mitochondrial respiratory chain I (*Owen et al., 2000*) and/or glycerophosphate dehydrogenase in the mitochondrial electron transport chain (*Madiraju et al., 2014*), to P0 *Tsc1fl/+;Tyrp1-Cre* and *Tsc1fl/fl;Tyrp1-Cre* mice. We found that 2DG treatment decreased the lactate level in *Tsc1fl/+;Tyrp1-Cre* mouse retinas while metformin increased this parameter (*Figure 7—figure supplement 1C*). Moreover, the lactate level in 2DG-treated *Tsc1fl/fl;Tyrp1-Cre* became similar to that of PBS-treated *Tsc1fl/+;Tyrp1-Cre* mouse retinas, suggesting that the lactate accumulation in *Tsc1fl/fl;Tyrp1-Cre* was caused by enhanced glycolysis. We also found that not only total numbers of CldU-labeled proliferating RPCs but also those that had reentered the cell cycle (assessed by monitoring the incorporation of IdU, such that the cells were CldU;IdU-positive) were decreased significantly in 2DG-treated *Tsc1fl/fl;Tyrp1-Cre* mouse retinas in comparison to the numbers in the PBS- and metformin-injected groups and became similar numbers observed in *Tsc1fl/+;Tyrp1-Cre* littermate mouse retinas (*Figure 7H-J*). These results suggest that *Tsc1*-deficient RPCs depend on glycolysis for their accelerated cell cycle progression.

Collectively, our results suggest that mTORC1 increases the cellular Hif1a protein level through both translational and post-translational mechanisms in the RPCs of *Tsc1fl/fl;Tyrp1-Cre* mouse retinas by generating a hypoxic condition. This elevated Hif1a facilitates the expression of glycolytic enzymes to supply the ATP necessary for RPC proliferation (*Figure 7K*). This enables the RPCs to expand rapidly to exceed their mitotic division limit prior to producing the MG, which degenerate from senescence-associated cell death in mature *Tsc1fl/fl;Tyrp1-Cre* mouse retina.

## Discussion

The growth and differentiation of vertebrate nervous tissues are tightly controlled in specific spatial and temporal manners to ensure the proper formation of their unique three-dimensional structures. During the growth of the double-layered optic cup of vertebrates, cell proliferation is significantly faster at the inner cup layer compared to the outer cup layer (*Moon et al., 2018*). Here, we detected mTORC1 activity only in the inner optic cup layer (*Figure 1—figure supplement 1*), suggesting that mTORC1 may involve in the spatially differentiated growth of the optic neuroepithelial continuum. In support of this, hamartomatous malformation of the CB and iris were reported in human TSC patients and in mice lacking *Tsc1* throughout the optic cup (*Eagle et al., 2000*; *Hägglund et al., 2017*; *Milea and Burillon, 1997*). We also found CB/iris malformation in *Tsc1fl/fl;Rax-Cre* mice, in which *Tsc1* was lost throughout the optic neuroepithelium, and in *Tsc1fl/fl;Tyrp1-Cre* mice, in which *Tsc1* was lost in the RPE, CM, and peripheral retinal clones. In contrast, we did not observe CB/iris defects in mice lacking

*Tsc1* only in the retina (i.e., *Tsc1*$^{fl/fl}$;*Chx10-Cre* mice) and RPE (i.e., *Tsc1*$^{fl/fl}$;*Mlana-Cre* mice) (**Figure 3A**). This suggests that TSC plays an essential role in the CM to generate the CB and iris by suppressing the growth of the areas. However, the deletion of *Tsc2* from the RPE and CM of *Tsc2*$^{fl/fl}$;*Tyrp1-Cre* mice did not cause CB/iris malformations, although the mice shared the retinal phenotypes of *Tsc1*$^{fl/fl}$;*Tyrp1-Cre* mice (**Figure 1—figure supplement 4**). These results suggest that Tsc1 regulates CB and iris development independent of the mTORC1 pathway. Given reported findings that Tsc1 functions in the tumor growth factor β (TGFβ)-Smad pathway (**Thien et al., 2015**) and the activation of this pathway in mouse CB (**Srinivasan et al., 1998**), we speculate that Tsc1 might contribute to CB and iris development via the TGFβ-Smad pathway. In parallel, we show Tsc1 also cooperates with Tsc2 to suppress mTORC1 in CM RPCs, whose cell division rate should be maintained at low to avoid of precocious arrival of the RPCs to the division limit. Collectively, these results suggest that Tsc1 is a multifunctional regulator of vertebrate eye development.

Our data suggest that the deletion of *Tsc1* in a small RPC subset derived from the CM can cause the degenerative phenotypes in the mature retina (**Figure 1**). The CM RPCs in mouse retina were found to proliferate less robustly than the majority RPCs in the central retina (**Belanger et al., 2017**; **Marcucci et al., 2016**), so did the RPCs in lower vertebrate CMZ (**Harris and Perron, 1998**). However, the CM RPCs could have a competitive advantage for clonal expansion over the majority wild-type RPCs, which divide and expand slowly, when their cell cycle progression is accelerated by losing *Tsc1* (**Figure 4—figure supplement 4**). This made the *Tsc1*-deficient RPCs divide exceeding the division limit and cause the degenerative phenotypes in *Tsc1*$^{fl/fl}$;*Tyrp1-Cre* mouse retinas. The competitive hyperexpansion of a *Tsc1*-deficient RPC exceeding the division limit did not occur in *Tsc1*$^{fl/fl}$;*Rax-Cre* and *Tsc1*$^{fl/fl}$;*Chx10-Cre* mouse retinas, since its neighboring RPCs also lose *Tsc1* and expand as fast as the RPC (**Figures 4C and 5A**; **Figure 4—figure supplement 4**). Therefore, *Tsc1*-deficient RPCs should be present in a minor population to have a competitive advantage in the clonal expansion.

The competitive clonal expansion could also occur in other conditions, where two RPC populations expand at different speeds. The *Rptor*$^{fl/fl}$;*Tyrp1-Cre* mouse retina, which is composed of relatively fast dividing wild-type RPCs and slowly dividing (or cell cycle arrested) *Rptor*-deficient RPCs, could be an example. However, wild-type RPCs in the *Rptor*$^{fl/fl}$;*Tyrp1-Cre* mouse retina did not overproliferate even though their neighboring *Rptor*-deficient RPCs stopped expanding their clones (**Figure 5F–J**). Therefore, the overproliferation might be a specific feature of *Tsc1*-deficient RPC but not a relative property obtained by the division differences.

The MG are among the last-born retinal cell types, arising around the first postnatal week in mice (**Cepko, 2014**; **Hoang et al., 2020**). Therefore, given the specific temporal sequence of retinal histogenesis, RPCs producing MG are likely to be mitotically older than those producing other retinal cell types. Since *Tsc1*-deficient RPCs overproliferate during the embryonic and early postnatal periods (**Figure 4E**), the mitotic ages of *Tsc1*-deficient RPCs in *Tsc1*$^{fl/fl}$;*Tyrp1-Cre* mice should be far greater than those of wild-type RPCs when they are producing MG. Consequently, the MG born from *Tsc1*-deficient RPCs might be degenerated due to aging-related changes (**Figure 5D and E**). However, we found little MG degeneration or rosette formation in the retinas of old (2 years) wild-type C57BL/6 J mice (data not shown), suggesting that the phenotypes of *Tsc1*$^{fl/fl}$;*Tyrp1-Cre* and *Tsc2*$^{fl/fl}$;*Tyrp1-Cre* mouse retinas are not simple aging-related changes.

Among the last-born retinal cell types, only MG exhibited significant degeneration in *Tsc1*$^{fl/fl}$;*Tyrp1-Cre* mouse retina (**Figure 1D**). The MG might be in less terminal stage than the other last-born types (i.e., rPR and BC) in terms of differentiation, since they can resume cell cycle to regenerate neurons in the injured cold-blooded vertebrate retinas (**Lahne et al., 2020**). Mouse MG could also divide after the injury, if histone deacetylase inhibitor is provided after viral expression of a proneural transcription factor *achaete-scute homolog 1* (**Jorstad et al., 2017**). Furthermore, given the fact that developing cells are more sensitive than terminally differentiated cells to cell death (**Fuchs and Steller, 2011**; **Vaux and Korsmeyer, 1999**), MG are likely more sensitive to the cell death than rPR and BC. However, the mechanisms underlying the differential apoptotic sensitivity of these last-born retinal cell types should be investigated in future studies.

The factors responsible for inducing the death of *Tsc1*-deficient RPCs and/or MG in *Tsc1*$^{fl/fl}$;*Tyrp1-Cre* mouse retinas remain unclear. The repetitive division of RPCs might exploit telomeres beyond the critical limit and result in mitotic catastrophe, leading to senescence-associated cell death (**Cleal and Baird, 2020**). However, telomere shortening did not appear to be a direct cause of MG degeneration

in the mouse retinas, since it was not suppressed by *Tert* overexpression (*Figure 5—figure supplement 2*). Furthermore, unlike the strong relationship between telomere shortening and senescence in human cells, telomere shortening in mouse cells is rarely associated with deleterious effects because the mouse telomere is 5–10 times longer than the human telomere (*Calado and Dumitriu, 2013*). Therefore, the senescence observed in *Tsc1^fl/fl^;Tyrp1-Cre* mouse retina might be caused by other cellular alterations associated with the RPC overproliferation.

The activation of mTORC1 by *Tsc1* deletion was also shown to cause premature aging in mouse intestinal stem cells (ISCs) (*He et al., 2020*). The aging of *Tsc1*-deficient ISCs was reported to mediate the stress-induced p38 mitogen-activated protein kinase (MAPK) pathway, which activates p53 to block ISC proliferation. We also observed the activation (i.e., phosphorylation) of p38 MAPK in all four *Tsc1-cko* mouse retinas (data not shown), whereas senescence occurred only in *Tsc1^fl/fl^;Tyrp1-Cre* mouse retinas (*Figure 5C*). The activation of p53 did not likely cause degeneration of *Tsc1*-deficient RPCs and/or MG, either, since *Trp53^fl/fl^;Tsc1^fl/fl^;Tyrp1-Cre* mice still exhibited MG degeneration and retinal rosette formation (data not shown). Therefore, the activations of p38 MAPK and p53 may represent a cellular stress condition induced by mTORC1 hyperactivation, which increases cellular metabolism to elevate reactive oxygen species as byproducts (*Ben-Sahra and Manning, 2017*), but do not appear to trigger the senescence-associated cell death in *Tsc1^fl/fl^;Tyrp1-Cre* mouse retinas. A previous study further showed that intestinal epithelial cells (IECs) derived from *Tsc1*-deficient ISCs were degenerated via receptor interacting protein kinase 3 (Ripk3)-mediated necroptosis (*Xie et al., 2020*). However, in contrast to the critical roles of Ripk3 in the degeneration of *Tsc1*-deficient IECs, the MG degeneration in *Tsc1^fl/fl^;Tyrp1-Cre* mice was not blocked by concomitant deletion of *Ripk3* (data not shown). These results suggest that the mechanism of MG degeneration in *Tsc1^fl/fl^;Tyrp1-Cre* mice might be different from the stress-induced aging and degeneration of IECs, and thus further investigation is warranted.

Embryonic and neonatal mouse retinas receive oxygen that diffuses from the distant hyaloid vessels in the vitreous and the choroid vessels across the RPE, whereas the mature mouse retina has close access to the oxygen released from intraretinal blood vessels (*Lutty and McLeod, 2018*; *Saint-Geniez and D'amore, 2004*). RPCs in the inner embryonic mouse retinas are farthest from the vessels, and are likely to exist under more hypoxic conditions than cells adjacent to the vitreous and RPE. The inner RPCs therefore might adapt to use glycolysis ATP synthesis than oxygen-dependent mitochondrial ATP synthesis for proper proliferation, as has been suggested for *Xenopus* retina (*Agathocleous et al., 2012*). Furthermore, *Tsc1*-deficient RPCs might spend more ATP for fast growth and division, compared to wild-type RPCs (*Figure 7—figure supplement 1A*), making their mitochondria utilize more oxygen to supply ATP. This is expected to worsen the hypoxia of *Tsc1^fl/fl^;Tyrp1-Cre* mouse RPCs and thereby compromise their mitochondrial ATP production eventually. In this situation, glycolytic ATP synthesis compensates for the ATP shortage in hypoxic *Tsc1*-deficient RPCs through the mTORC1-induced enhanced translation and stabilization of Hif1a, which, in turn, increases the expression of glycolytic enzymes (*Figure 7D–F*). The glycolytic gene expression is therefore decreased in the absence of Hif1a, such that ATP cannot be produced at a level sufficient to support mTORC1-induced cell proliferation and growth. This might decelerate the growth and proliferation of *Tsc1*-deficient RPCs and normalize the speed of retinal development.

# Materials and methods

## Key resources table

| Reagent type (species) or resource | Designation | Source or reference | Identifiers | Additional information |
|---|---|---|---|---|
| Antibody | Anti-Aquaporin(AQP1) (Mouse monoclonal) | Novus Biologicals | NB600-749 | IHC (1:200) |
| Antibody | Anti-beta-actin (Rabbit polyclonal) | Santa Cruz Biotechnology | sc-1616 | WB (1:1000) |
| Antibody | Anti-beta-catenin (Rabbit polyclonal) | Cell Signaling Technology | 9562 | IHC (1:200) |
| Antibody | Anti-beta-galactosidase (Mouse monoclonal) | Developmental Studies Hybridoma Bank (DSHB) | JIE7 | WB (1:500) IHC (1:100) |

*Continued on next page*

*Continued*

| Reagent type (species) or resource | Designation | Source or reference | Identifiers | Additional information |
|---|---|---|---|---|
| Antibody | Anti-BrdU(CldU) (Rat monoclonal) | Novus Biologicals | NB500-169 | IHC (1:200) |
| Antibody | Anti-BrdU(IdU) (Mouse monoclonal) | EXBIO | 11–286 C100 | IHC (1:200) |
| Antibody | Anti-Brn3b (Rabbit polyclonal) | Santa Cruz Biotechnology | sc-31989 | IHC (1:200) |
| Antibody | Anti-Calbindin (Rabbit polyclonal) | Swant Inc. | CB-38 | IHC (1:200) |
| Antibody | Anti-Cdo (Goat polyclonal) | R&D Systems | AF2429 | IHC (1:200) |
| Antibody | Anti-c-myc (Mouse monoclonal) | Santa Cruz Biotechnology | sc40 | WB (1:1000) |
| Antibody | Anti-Cleaved caspase-3 (Rabbit polyclonal) | Cell Signaling Technology | 9661 | IHC (1:200) |
| Antibody | Anti-Ezrin (Mouse monoclonal) | Invitrogen Biotechnology | 35–7300 | IHC (1:200) |
| Antibody | Anti-GFAP (Rabbit polyclonal) | Abcam | ab48050 | IHC (1:200) |
| Antibody | Anti-GFP (Chicken polyclonal) | Abcam | ab13970 | IHC (1:2000) |
| Antibody | Anti-Glutamine synthase (Rabbit polyclonal) | Sigma-Aldrich | G-2781 | IHC (1:200) |
| Antibody | Anti-Hif1a (Mouse monoclonal) | R&D Systems | MAB1536 | WB (1:1000) |
| Antibody | Anti-Hif2$\alpha$/Epas1 (Rabbit polyclonal) | Novus Biologicals | NB100-122 | WB (1:1000) |
| Antibody | Anti-Hk2 (Rabbit polyclonal) | Cell Signaling Technology | 2867 | WB 1:1,000 IHC (1:200) |
| Antibody | Anti-Hydroxy-Hif1alpha (Rabbit polyclonal) | Cell Signaling Technology | 3434 | WB (1:1000) |
| Antibody | Anti-M-opsin | Merck Millipore | AB5405 | IHC (1:200) |
| Antibody | Anti-Otx2 (Rabbit polyclonal) | Abcam | ab25985 | IHC (1:200) |
| Antibody | Anti-Otx2 (Rabbit polyclonal) | Abcam | ab183951 | IHC (1:200) |
| Antibody | Anti-Otx2 (Goat polyclonal) | R&D Systems | AF1979-SP | IHC (1:200) |
| Antibody | Anti-p21/Cip1 (Mouse monoclonal) | Santa Cruz Biotechnology | SC817 | WB (1:1000) IHC (1:200) |
| Antibody | Anti-p53 (Mouse monoclonal) | Merck Millipore | OP03-100 | WB (1:1000) IHC (1:200) |
| Antibody | Anti-Pax6 (Rabbit polyclonal) | Covance | PRB-278P | IHC (1:200) |
| Antibody | Anti-Phospho Smad1/5(S463/465) (Rabbit polyclonal) | Invitrogen Biotechnology | 700047 | IHC (1:200) |
| Antibody | Anti-phospho Smad2(ser465/467) (Rabbit polyclonal) - | Cell Signaling Technology | 18338 | IHC (1:200) |
| Antibody | Anti-phospho-Histone H3(S10) (pH3; Rabbit polyclonal) | Merck Millipore | 04–1093 | IHC (1:200) |
| Antibody | Anti-phospho-S6(S235/236) (pS6; Rabbit polyclonal) | Cell Signaling Technology | 2211 | WB (1:1000) IHC (1:200) |

*Continued on next page*

*Continued*

| Reagent type (species) or resource | Designation | Source or reference | Identifiers | Additional information |
|---|---|---|---|---|
| Antibody | Anti-Psmb8 (Mouse monoclonal) | Enzo Life Science | BML-PW8845 | WB (1:1000) |
| Antibody | Anti-Psmb9 (Mouse monoclonal) | Santa Cruz Biotechnology | sc-373996 | WB (1:1000) |
| Antibody | Anti-Psmb10 (Rabbit polyclonal) | Abcam | ab1183506 | WB (1:1000) |
| Antibody | Anti-Raptor (Rabbit polyclonal) | Cell Signaling Technology | 2280 | WB (1:1000) |
| Antibody | Anti-PKM2 (Rabbit polyclonal) | Abgent | ap7173d | WB (1:1000) |
| Antibody | Anti-PKM2 (Rabbit polyclonal) | Cell Signaling Technology | 4053 | IHC (1:200) |
| Antibody | Anti-Recoverin (Rabbit polyclonal) | Chemicon | AB5585 | WB (1:1000) |
| Antibody | Anti-RPE65 (Mouse monoclonal) | Abcam | ab13826 | WB (1:1000) |
| Antibody | Anti-Rhodopsin (Mouse monoclonal) | Chemicon | MAB5356 | IHC (1:200) |
| Antibody | Anti-S6 (Mouse monoclonal) | Cell Signaling Technology | 2317 | WB (1:1000) |
| Antibody | Anti-Tom20 (Rabbit polyclonal) | Santa Cruz Biotechnology | sc-11415 | WB (1:1000) |
| Antibody | Anti-Tsc1 (Rabbit polyclonal) | Cell Signaling Technology | 4906 | WB (1:1000) |
| Antibody | Anti-Tsc2 (Rabbit polyclonal) | Cell Signaling Technology | 3612 | WB (1:1000) |
| Antibody | Anti-Tubulin-ßIII (Tuj1; Mouse monoclonal) | Covance | MMS-435P | IHC (1:200) |
| Antibody | Anti-Vsx2 (Mouse monoclonal) | Santa Cruz Biotechnology | sc365519 | WB (1:1000) IHC (1:200) |
| Genetic reagent (*Mus musculus*) | *B6.129-Hif1a^tm3Rsjo^/J* | Jackson Laboratory | 007561 | *Hif1a^fl/fl^* |
| Genetic reagent (*Mus musculus*) | B6.Cg-*Rptor^tm1.1Dmsa^*/J | Jackson Laboratory | 013188 | *Rptor^fl/fl^* |
| Genetic reagent (*Mus musculus*) | *Tsc1^tm1Djk^*/J | Jackson Laboratory | 005680 | *Tsc1^fl/fl^* |
| Genetic reagent (*Mus musculus*) | *Tsc2^tm2.1Djk^*/Mmjax | Jackson Laboratory | 37154-JAX | *Tsc2^fl/fl^* |
| Genetic reagent (*Mus musculus*) | *CAG-loxP-3xpolyA-loxP-mTERT-IRES-Hygro-r* | **Hidema et al., 2016** | | *dTert* |
| Genetic reagent (*Mus musculus*) | *Tg(Tyrp1-cre)1Ipc* | **Mori et al., 2002** | | *Tyrp1-Cre* |
| Genetic reagent (*Mus musculus*) | *Tg(Mlana-cre)5Bee* | **Aydin and Beermann, 2011** | | *Mlana-Cre* |
| Genetic reagent (*Mus musculus*) | *Tg(Chx10-EGFP/cre,-ALPP)2Clc* | **Rowan and Cepko, 2004** | | *Chx10-Cre* |
| Genetic reagent (*Mus musculus*) | *Tg(Pax6-cre,GFP)2Pgr* | **Marquardt et al., 2001** | | *Pax6-αCre* |
| Genetic reagent (*Mus musculus*) | *Tg(Rax-cre)1Zkoz* | **Klimova et al., 2013** | | *Rax-Cre* |
| Genetic reagent (*Mus musculus*) | *Tg(Slc1a3-cre/ERT)1Nat/J* | Jackson Laboratory | 012586 | *Slc1a3-CreERT2* |

*Continued*

| Reagent type (species) or resource | Designation | Source or reference | Identifiers | Additional information |
|---|---|---|---|---|
| Genetic reagent (*Mus musculus*) | B6.129 × 1-*Gt(ROSA)26Sor*$^{tm1(EYFP)Cos}$/J | Jackson Laboratory | 006148 | *R26*$^{EYFP}$ |
| Genetic reagent (*Mus musculus*) | B6.Cg-*Gt(ROSA)26Sor*$^{tm14(CAG-tdTomato)Hze}$/J | Jackson Laboratory | 007914 | *R26*$^{tdTomato}$ |
| Genetic reagent (*Mus musculus*) | *Tg(Tyrp1-cre/ERT2)1Jwk* | This paper | | *Tyrp1-CreERT2* |
| Genetic reagent (*Mus musculus*) | B6-*Psmb8em1hwl Psmb9em1hwl/Korl* | This paper | | *Psmb8*$^{-/-}$,*9*$^{-/-}$ |
| Genetic reagent (*Mus musculus*) | B6-*Psmb10em1Jwk* | This paper | | *Psmb10*$^{-/-}$ |
| Chemical compound, drug | Phalloidin-AlexaFluor 647 | Abcam | ab176759 | 1:200 |
| Chemical compound, drug | Hoechst 33,342 | Invitrogen | H1399 | 1:1,000 |
| Chemical compound, drug | Click-iT EdU Cell Proliferation Kit for Imaging, Alexa Fluor 647 dye | Invitrogen | C10340 | |
| Chemical compound, drug | 2-Deoxy-D-glucose | Sigma-Aldrich | D6134 | |
| Chemical compound, drug | Metformin hydrochloride | Sigma-Aldrich | PHR1084 | |
| Chemical compound, drug | 2-Chloro-5-deoxyuridine (CldU) | Sigma-Aldrich | C6891 | |
| Chemical compound, drug | 5-Iodo-2'-deoxyuridine | Sigma-Aldrich | I7125 | |
| Chemical compound, drug | Methyl cellulose | Sigma-Aldrich | M7140 | |
| Chemical compound, drug | L-Glutamine | GIBCO | 25030–081 | |
| Chemical compound, drug | B27 supplement | GIBCO | 17504–044 | |
| Chemical compound, drug | N2 supplement | GIBCO | 17502–048 | |
| Chemical compound, drug | Heparin | Millipore | M535142 | |
| Chemical compound, drug | Protease inhibitor | Millipore | M535142 | |
| Peptide, recombinant protein | Dnase I | Sigma-Aldrich | DN25-100mg | |
| Peptide, recombinant protein | Epidermal growth factor (EGF) | Sigma-Aldrich | E4127 | |
| Peptide, recombinant protein | Fibroblast growth factor 2 (hBFGF) | Sigma-Aldrich | F0291 | |
| Commercial assay or kit | ENLITEN ATP Assay System | Promega | FF2000 | |
| Commercial assay or kit | Lactate-Glo Assay | Promega | J5021 | |
| Commercial assay or kit | SuperSignal West Pico plus chemiluminescent substrate | Thermo Scientific | 34580 | |
| Commercial assay or kit | SuperSignal West Femto plus chemiluminescent substrate | Thermo Scientific | 34095 | |
| Commercial assay or kit | In situ Cell death Detection Kit, TMR-red | Roche | 12156792910 | |
| Commercial assay or kit | In situ Cell Death Detection Kit, Fluorescein (TUNEL green) | Roche | 11684 795910 | |
| Commercial assay or kit | Hypoxyprobe Kit | Hypoxyprobe | HP1-100kit | |
| Software, algorithm | Fluoview 4.0 | Olympus Corporation | N/A | |
| Software, algorithm | Imaris 9.3 | Bitplane | N/A | |
| Software, algorithm | GraphPad Prism v7.0 | GraphPad software | N/A | |
| Sequence-based reagent | *Actinß1*-forward | Bioneer | RT-qPCR primer | CTGGCTCCTAGCACCATGAAGAT |
| Sequence-based reagent | *Actinß1*-reverse | Bioneer | RT-qPCR primer | GGTGGACAGTGAGGCCAGGAT |

*Continued on next page*

*Continued*

| Reagent type (species) or resource | Designation | Source or reference | Identifiers | Additional information |
|---|---|---|---|---|
| Sequence-based reagent | *Pkm2*-forward | Bioneer | RT-qPCR primer | TCGCATGCAGCACCTGATT |
| Sequence-based reagent | *Pkm2*-reverse | Bioneer | RT-qPCR primer | CCTCGAATAGCTGCAAGTGGTA |
| Sequence-based reagent | *Hk2*-forward | Bioneer | RT-qPCR primer | TGATCGCCTGCTTATTCACGG |
| Sequence-based reagent | *Hk2*-reverse | Bioneer | RT-qPCR primer | AACCGCCTAGAAATCTCCAGA |
| Sequence-based reagent | *Hif1a*-forward | Bioneer | RT-qPCR primer | ACCTTCATCGGAAACTCCAAAG |
| Sequence-based reagent | *Hif1a*-reverse | Bioneer | RT-qPCR primer | CTGTTAGGCTGGGAAAAGTTAGG |
| Sequence-based reagent | *Psmb8*-forward | Bioneer | Genotyping primer | TTGGTACTGTGGCTTTCGCTTTC |
| Sequence-based reagent | *Psmb8*-reverse | Bioneer | Genotyping primer | ACACTCCTTCCTCTGTGCCACC |
| Sequence-based reagent | *Psmb9*-forward | Bioneer | Genotyping primer | GACCTTGAGTCGGTCACCTCC |
| Sequence-based reagent | *Psmb9*-reverse | Bioneer | Genotyping primer | CACTTAGGGCCACCAGCTTCC |
| Sequence-based reagent | *Psmb10*-forward | Bioneer | Genotyping primer | ACGCGAGTCACCCCA ATGTTT |
| Sequence-based reagent | *Psmb10*-reverse | Bioneer | Genotyping primer | CGCCACAACCGAATCGTTAGT |
| Sequence-based reagent | Psmb8-gRNA forward | Bioneer | CRISPR/Cas9 | TCGGGGGCAGCGGCCCGAGTGGG |
| Sequence-based reagent | Psmb8-gRNA reverse | Bioneer | CRISPR/Cas9 | CCAGGGCAGCCCACTCGGGCCGC |
| Sequence-based reagent | Psmb9-gRNA forward | Bioneer | CRISPR/Cas9 | GGAGTTTGACGGGGGTGTCGTGG |
| Sequence-based reagent | Psmb9-gRNA reverse | Bioneer | CRISPR/Cas9 | CCCACCACGACACCCCCGTCAAA |
| Sequence-based reagent | Psmb10-gRNA #3 forward | Bioneer | CRISPR/Cas9 | CACCGAACACGTCCTTCCGGGACTT |
| Sequence-based reagent | Psmb10-gRNA #3 reverse | Bioneer | CRISPR/Cas9 | AAACAAGTCCCGGAAGGACGTGTTC |
| Sequence-based reagent | Psmb10-gRNA #5 forward | Bioneer | CRISPR/Cas9 | CACCGCTGCCAGAGGAATGCGTCCT |
| Sequence-based reagent | Psmb10-gRNA #5 reverse | Bioneer | CRISPR/Cas9 | AAACAGGACGCATTCCTCTGGCAGC |

## Mice

Information of mouse strains used in the experiments is listed in Key resources table. *Psmb8*[-/-],*9*[-/-],*10*[-/-] triple knock-out (*Psmb-tko*) and *Tyrp1-CreERT2* mice were generated in this study. The *Psmb-tko* mice were generated using the CRISPR/Cas9 system, as it was described in a previous report (*Kim et al., 2017*). The genetic manipulations resulted in 25, 8, and 22 nucleotide (nt) deletions in mouse *Psmb8*, *Psmb9, and Psmb10* genes, respectively (*Figure 6—figure supplement 2*). The sequences of guide RNAs used for introducing the deletions are provided in Key resources table. The *Tyrp1-CreERT2* mice were generated following the same procedure reported previously (*Mori et al., 2002*). The transgenic mice were established by microinjection of the 7.3 kb NotI DNA segment of *Tyrp1-Cre*[ERT2] into *C57BL/6* J mouse embryos (two-cell stage). These injected embryonic cells were then injected into the inner cell mass of ICR mouse embryos. The tails of mice born from the surrogate mice were isolated for the genotyping with the primers listed in Key resources table. Two resulting F1 chimeric male mice carrying the transgene were crossed to *C57BL6/J* female mice to obtain an F2 generation with the heterozygous *Tyrp1-CreERT2* mice.

To generate the cko mice, the mice having the floxed (*fl*) target gene alleles were bred with the mice expressing various Cre recombinases. To tracing of the cells experienced the Cre-dependent recombination at target gene loci, the mice were crossed with the *R26*[EYFP] and *R26*[tdTom] mice, and the cells experienced the Cre-dependent recombination of target sites were visualized by the fluorescence of the reporters. Experiments using the mice were carried out according to the guidance of Institutional

Animal Care and Use Committee (IACUC) of KAIST (KA-2014–20). All mice used in this study were maintained in a specific pathogen-free facility of KAIST Laboratory Animal Resource Center.

## Cryosections and immunohistochemistry

Pregnant and postnatal mice were euthanized after the anesthesia by intraperitoneal injection of tribromoethanol (Avertin, Sigma). The euthanized postnatal mice were perfused with phosphate buffered saline (PBS, pH7.5) containing 0.1 % heparin (Millipore) and then with 4 % paraformaldehyde (PFA, Sigma) in PBS. The eyes were isolated from the mice for further fixation in 4 % PFA/PBS solution at 4 °C for 2 hr. The embryos were isolated from the pregnant mice and fixed in 4 % PFA/PBS solution at 4 °C for 4 hr. The eyes and embryonic heads were then transferred to 20 % sucrose/PBS solution for the incubation at 4 °C for 16 hr before embedding in the Tissue-Tek OCT compound (Sakura). Cryosections (10–14 μm) of the frozen embryonic heads and postnatal mouse retinas were obtained and stained with hematoxylin and eosin (H&E) solutions for histologic examinations.

For immunohistochemistry, the sections were incubated for 1 hr in a blocking solution containing 10 % normal donkey serum in PBS containing 0.2 % Triton X-100. The sections were incubated with the primary antibodies at 4 °C for 16 hr, and further stained with Alexa488-, Cy3-, or Alexa647-conjugated secondary antibodies (Jackson ImmunoResearch Laboratories) at room temperature (RT) for 1 hr. The antibodies are listed in Key resources table. Fluorescent images were obtained by confocal microscope (Fluoview FV1000 and FV3000; Olympus).

For the detection of proliferating cells in the embryos or neonatal mouse retinas, pregnant and neonatal mice were injected with 5-bromo-2'-deoxyuridine (EdU; 50 mg/kg, ThermoFisher). EdU was detected by incubating in Click-iT EdU Alexa Fluor 647 (Thermo Fisher Scientific) according to manufacturer's instructions. Alternatively, the mice were injected with 5-chloro-2'-deoxyuridine (CldU; 30 mg/kg, Sigma) and 5-iodo-2'-deoxyuridine (IdU; 30 mg/kg, Sigma) into their peritoneal cavity for the indicated time periods prior to sample collection. The eyes were isolated from the mice and the eye sections were post-fixed in 4 % PFA/PBS for 5 min, and washed three times with PBS with Triton X-100 0.2 %. The sections were then incubated with 2 N HCl for 30 min, neutralized with borate buffer (pH 8.0) for 5 min (three times, 10 min) at room, followed by rinses with PBS (three times, 10 min). The sections were then subjected to the immunohistochemistry procedures described above.

## Reverse transcription-quantitative polymerase chain reaction analysis

Total RNA from mouse retinas were isolated using easy-BLUE Total RNA Extraction Kit (iNtRON). About 1 μg of total RNA was reverse transcribed using SuperiorScript III Master Mix (Enzynomics) according to the manufacturer's protocol. Real-time quantitative polymerase chain reaction (PCR) (RT-qPCR) was performed with TOPreal qPCR 2 X PreMIX (Enzynomics). The RT-qPCR was performed at 95 °C for 10 min, followed by 50 cycles of 95 °C for 15 s and 60 °C for 15 s. Relative expression levels of gene of interest were calculated according to the $2^{-\Delta\Delta Ct}$ method against β-actin mRNA. Primer sequences are listed in Key resources table.

## Quantification of relative telomere lengths

Genomic DNA (gDNA) were extracted from mouse retinas by lysing the cells in DirectPCR Lysis Reagent (Viagen Biotech) supplemented with RNaseA (2 mg/ml, Bioneer) and proteinase-K (2 mg/ml, Roche), and then were purified further by the extraction with phenol-chloroform-isoamyl alcohol (25:24:1 (v/v); Sigma) followed by precipitation with 0.3 M (final) sodium acetate (pH 5.2) and 0.7 volume isopropanol. The lengths of telomeres in the gDNA dissolved in Tris-EDTA (pH 8.0) solution were then assessed by Relative mouse Telomere Length quantification qPCR assay kit (ScienCell Research Laboratories) according to the manufacturer's protocol.

## Western blot analysis

Total protein was extracted from mouse retinas with RIPA buffer (50 mM Tris [pH 8.0], 150 mM NaCl, 1.0% NP40, 0.5 % sodium deoxycholate, 0.1 % sodium dodecyl sulfate [SDS]) supplemented with the protease and phosphatase inhibitor cocktail (Roche). For separation of proteins, the cell lysates containing 30 μg proteins were analyzed by 8–15 % SDS-polyacrylamide gel electrophoresis (SDS-PAGE) prior to the transfer onto polyvinylidene difluoride membranes. The membranes were blocked with 5 % skim milk in Tris-buffered saline containing 0.1 % Tween-20 (TBST, pH 7.4) at RT for 1 hr.

The membranes were incubated at 4 °C for 16 hr with primary antibodies listed in Key resources table, washed with the TBST at RT three times (10 min each), and incubated with the horseradish peroxidase-conjugated secondary antibodies at RT for 1 hr. Reactive bands were detected using SuperSignal West Pico Chemiluminescent Substrate or SuperSignal West Femto Maximum Sensitivity Substrate (Thermo Fisher Scientific).

### FACS analysis

Mouse retinas were collected in 1 ml Hank's Balanced Salt Solution (HBSS; Life Technologies) after enucleating the lens from the eyes, and transferred into HBSS containing 0.1 % Trypsin and DNase I (100 µg/ml; Sigma) followed by the incubation at 37 °C for 30 min. Dissociated retinal cells were gently triturated in HBSS with 2 % FCS, filtered through a 70 µm strainer before FACS analysis. The R26tdTom-positive cells were then analyzed by the BD Fortessa analyzer (BD Biosciences) or collected by the BD FACSAria cell sorter (BD Biosciences).

### RPC neurosphere culture

The retinas of E13 mouse embryos were isolated and incubated in a dissociation solution (DMEM containing 0.1 % Trypsin and DNase I [100 µg/ml; Sigma]) at 37 °C for 30 min. The aggregates were further dissociated into single cells using Accutase and passed through a 70 µm cell strainer. The R26tdTom-positive cells were then collected by FACS for subsequent culture in DMEM/F12 medium containing 0.9% (w/v) methylcellulose matrix (Sigma), N2 supplement (GIBCO), B27(GIBCO), 2 mM L-glutamine (GIBCO), 100 U/ml penicillin, and 100 µg/ml streptomycin (GIBCO) supplemented with 10 ng/ml fibroblast growth factor 2 (Sigma) and 20 ng/ml epidermal growth factor (Sigma). Then the cells were cultured for 1–4 weeks and the number of primary neurospheres was counted every 3 days with microscopic observation.

### ATP assay

ATP concentration was measured using the ENLIGHTEN ATP assay system (Promega). In brief, mouse retinas were homogenized in a buffer (0.25 M sucrose and 10 mM HEPES-NaOH, pH 7.4) and centrifuged at 3000 rpm at 4 °C for 10 min. The supernatant (200 µl) was added to an equal volume of 10 % trichloroacetic acid (Sigma) and centrifuged at 10,000 rpm at 4 °C for 10 min. After centrifugation, 300 µl of the supernatant was added with 200 µl of neutralization buffer (1 M Tris-acetate buffer, pH 7.5) and then was diluted 30-folds in deionized water prior to the measurement of luminescence using the MICROLUMAT LB96P Reader (Berthold).

### Lactate assay

Lactate concentration was measured using the Lactate-Glo Assay Kit (Promega) according to the manufacturer's protocol. Briefly, 1 mg of the retina were homogenized in 400 µl of homogenization buffer (50 mM Tris, pH 7.5) with 50 µl of inactivation solution (0.6 N HCL) and immediately added to 50 µl of neutralization buffer. Before recording luminescence, 50 µl of samples were transferred into a white 96-well plate and added 50 µl of lactate detection reagent to the well, mix for 60 s, and incubate at RT for 1 hr. Luminescence was recorded with the MICROLUMAT LB96P Reader (Berthold).

### Statistical analysis

Statistical analysis was performed by Prism Software (GraphPad) measurement tools. Data from statistical analysis are presented as the mean ± SD. Student's t test was used to determine the significant difference between two genotypes and one-way ANOVA with Tukey's post-test used to determine the significant differences among multiple groups. p-Values were calculated using a two-tailed unpaired $t$-test. $p < 0.05$ was considered statistically significant. $*p < 0.05$; $**p < 0.01$; $***p < 0.005$; $****p < 0.001$.

## Acknowledgements

This work was supported by the National Research Foundation of Korea (NRF) grants funded by Korean Ministry of Science and ICT (MSIT) (2017R1A2B3002862 and 2018R1A5A1024261; JWK); the grant funded by Samsung Foundation of Science and Technology (SSTF-BA1802-10; JWK); and the grant of Czech Science Foundation (21–27364S).

## Additional information

### Funding

| Funder | Grant reference number | Author |
|---|---|---|
| National Research Foundation of Korea | 2017R1A2B3002862 | Jin Woo Kim |
| National Research Foundation of Korea | 2018R1A5A1024261 | Jin Woo Kim |
| Samsung Science and Technology Foundation | SSTF-BA1802-10 | Jin Woo Kim |
| Czech Science Foundation | 21-27364S | Zbynek Kozmik |

The funders had no role in study design, data collection and interpretation, or the decision to submit the work for publication.

### Author contributions

Soyeon Lim, Conceptualization, Data curation, Formal analysis, Investigation, Methodology, Writing – original draft; You-Joung Kim, Formal analysis, Investigation, Methodology; Sooyeon Park, Ji-heon Choi, Young Hoon Sung, Han-Woong Lee, Methodology, Resources; Katsuhiko Nishimori, Zbynek Kozmik, Resources; Jin Woo Kim, Conceptualization, Data curation, Funding acquisition, Project administration, Supervision, Writing – original draft, Writing – review and editing

### Author ORCIDs

Ji-heon Choi http://orcid.org/0000-0001-9204-1755
Han-Woong Lee http://orcid.org/0000-0001-9515-3605
Jin Woo Kim http://orcid.org/0000-0003-0767-1918

### Ethics

Experiments using the mice were carried out according to the guidance of Institutional Animal Care and Use Committee (IACUC) of KAIST (KA-2014-20).

### Decision letter and Author response

Decision letter https://doi.org/10.7554/eLife.70079.sa1
Author response https://doi.org/10.7554/eLife.70079.sa2

## Additional files

### Supplementary files

• Transparent reporting form

• Source data 1. Uncropped Western blot (WB) images for those used in Figures 5C and I–7E are provided.

• Source data 2. Uncropped Western blot (WB) images for those used in Figure 1—figure supplement 2B, Figure 5—figure supplement 3A, Figure 5—figure supplement 3A, Figure 5—figure supplement 3B, Figure 6—figure supplement 2B are provided.

• Source data 3. Uncropped WB scan images for Main Figures.

• Source data 4. Uncropped WB scan images for figure supplements 1.

• Source data 5. Uncropped WB scan images for figure supplements 2.

### Data availability

All data generated or analysed during this study are included in the manuscript and supporting file; Source Data files have been provided for Figures 1 and 2.

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
