## [Editor Report]

Using a broad range of genetic and biochemical strategies, this study shows that hyperactivation of mTOR in retinal progenitors induces hyperproliferation and thus prematurely exhaust their mitotic capacity. They also show that these effects are related to Hif1alpha activation and metabolism. The study will be of considerably interest beyond field of retinal development, as it clearly links cell metabolism to tissue growth and perhaps competition.

---

## [Decision Letter]

Thank you for submitting your article "mTORC1-induced retinal progenitor cell over proliferation leads to accelerated mitotic aging and degeneration of descendent Müller glia" for consideration by *eLife*. Your article has been reviewed by 3 peer reviewers, and the evaluation has been overseen by a Reviewing Editor and Carlos Isales as the Senior Editor. The reviewers have opted to remain anonymous.

Essential revisions:

There is general agreement that the authors report a fascinating phenotype. However, there are also several concerns that are listed in the individual reviewers' reports. The common and most pressing unresolved issues that the authors should experimentally address are the following:

1) Is the Trp1-Cre line somehow contributing to the phenotype? This could be solved by the use of theTrp1-CreERT2 line or the injection so AAV-Cre. Extensive discussion of this issue is provided in the individual reviewers' assessment of the manuscript

2) Can the results be interpreted in terms of cell competition? The study does not provide sufficient support to the statement that telomere shortening is a "cell autonomous" effect.

*Reviewer #1:*

This paper exposes the effects of inactivating mTOR in the developing retina. The authors make the surprising discovery that multiple deficits are observed when mTOR is deleted in a fraction of retinal cells, but not when it is knocked out in most/all retinal cells. They tie these context specific deficits to aberrant Hif1alpha activity and its effects on metabolism.

Strengths:

Fascinating phenotype, uniquely observed in the context of mixed Tsc1 knockout (KO) and wild type (WT) cells. The authors performed extensive genetics to demonstrate the unique link to this context

Extensive characterization demonstrates telemore shortening, mitotic exhaustion and senescence associated with the phenotypic defects.

Additional genetics, biochemistry and phenotyping show, convincingly, that the effects depend in part on the proteasome and, more so, on the ability of Hif1alpha to engage glycolytic metabolism.

Weaknesses:

The authors did not consider the possibility that the phenotype reflects cell competition between fitter mutant and weaker normal cells i.e. that the latter are critical for the phenotype.

Comments on the author's conclusions:

The authors show convincingly that the effects are due to mitotic exhaustion and senescence, and that they are dependent on Hif1alpha. The conclusion that the effect is "cell autonomous" seems unjustified, as the genetic data suggests that normal cells may be essential for effects in the mutant population.

Potential Impact:

The paper will be of considerably interest, not only to those in the field of retinal development, but to those studying the link between mTOR, metabolism and tissue growth and (despite it not being raised in the current version) to those studying cell competition.

The authors show that deleting Tsc1 in the ciliary margin(CM)/RPE with TRP1-Cre stimulates expansion of these cells, and secondary retinal defects, such as rosette formation. Interestingly, deleting Tsc1 with multiple other retinal Cre strategies did not have that effect (GLAST-Cre/Muller; Chx10-Cre/most retinal progenitors; MART1-Cre/RPE, a-Cre/peripheral retina, mRx-Cre/retina+CM), and even combining Chx10-Cre + MART1-Cre did not mimic the effect seen with TRP1-Cre. These data suggest the cause is linked to the expansion of minority Tsc1 null progenitors, and not the absence of Tsc1 in retinal cell types per se, which is a unique and fascinating, while also puzzling, finding. Quantification comparing het and homozygous mice revealed that, indeed, the biggest increase in KO cells is seen in with the TRP1-Cre mice, with some increase in α-Cre, indicating that it is likely that KO cells are out-packing wild type cells in models where there are a mixture of mutant and wt genotype progenitors. Cultured neurosphere studies showed that KO cells generate larger spheres than wt cells, but degenerate earlier. Studies with P30 retina showed that TRP-Cre KO, but not other strains, exhibit telomere shortening and senescence. In the contrary experiment, inhibiting mTOR by knocking out floxed Raptor lengthened telomeres specifically in the TRP1-Cre context. mTOR activates the proteasome, and deleting proteasome subunits partially rescued retinal defects. mTOR can induce Hif1alpha, and indeed this protein was induced as expected, and KO rescued cellular and telomere defects in the mTRP1-Cre Tsc1 KO retina. Additional studies confirm accelerated cell cycle progression, ATP production, glycolysis, induction of glycolytic enzyme mRNA/protein and dependency of both on Hif1alpha.

– It is interesting, but overlooked in the paper, that the telomere length in KO cells is not intrinsically shorter in KO cells (e.g. not in mRx-Cre KO or CHx10-Cre KO retina), but only in retinas with mixed wt and KO cells (significant in TRP1-Cre KO, and n.s. in α-Cre but mean is lower). That finding suggests that the telomere shortening may be dependent on the presence of neighboring WT cells. This could, for example, reflect "cell competition" between KO and WT cells, in which the elevated rates of division is due to greater fitness of KO cells, that is stimulated by their weaker neighbors. Indeed, mTOR signaling has been linked to cell competition in mammals: PMID: 29720666; PMID: 33652024. It should be feasible to assess this notion by isolating early stage KO cells and assessing whether their division rate and/or telomere attrition is boosted specifically in the presence of WT cells relative to pure KO cultures.

– After showing telomere shortening in the KO but not wt cells from the same TRP1-Cre retina, the papers states that this phenomenon must be "cell autonomous". If that is the case, why is there no telomere shortening in other strains in which the entire/most of the retia is KO? As noted above, it appears that the shortening may depend on WT neighbors, in which case the term "cell autonomous" would be inappropriate.

– The same is true of other biochemical phenotypes that are unique to the TRP1-Cre retina, notably senescence. It's feasible, therefore, that the entire basis for the phenotypic specificity in the TRP1-Cre retina is dependent on mutant/normal cell interaction. The authors should attempt assays to validate or rule out that possibility. In addition to the cell mixing studies above, it should also be feasible to perform clonal studies in explants or in vivo using retroviral Cre delivery. It would, for example, be feasible to at least assess whether the resulting clones contain senescent cells (e.g. SA-betagal-positive staining).

– The authors show that expressing Tert does not rescue elevated cell death in the TRP1-Cre KO retina. They conclude on page 12: "This suggests that telomere shortening is unlikely to be the cause of MG degeneration in Tsc1fl/fl;TRP1-Cre mice". They would need to demonstrate that telomere length is rescued to make that claim, otherwise they should modify to state: "This may mean either that Tert expression is insufficient to rescue telomere length in this context, or that telomere shortening does not cause MG degeneration"

Overall, the paper is really interesting, but the authors have overlooked an explanation for their results that should be addressed, given the prominent literature on cell competition, and its link to mTOR.

*Reviewer #2:*

The manuscript titled, "mTORC1-induced retinal progenitor cell overproliferation leads to accelerated mitotic aging and degeneration of descendent Müller glia" by Lim et al., reports several conditional knockouts of Tsc1 within a variety of ocular compartments. The authors conclude that loss of Tsc1 within a subset of retinal progenitor cells (RPCs) leads to overactivation of mTORC1 and a consequential acceleration of cell cycle progression. They further conclude that this enhanced proliferation leads to an exhaustion of the mitotic capacity of these RPCs that go on to differentiate into Müller glia (MG) that subsequently degenerate due to premature "aging". Finally, they implicate Hif1a-mediated glycolytic gene expression as being required for this phenotype. The authors employ an impressive battery of Cre lines and floxed alleles aimed at precisely defining the cellular origin of the Trp1-Cre dependent Tsc1 CKO phenotype as well as its molecular mechanism. The majority of the data are sound and reflect strict adherence to scientific rigor.

From the data, they make the following main conclusions:

1. Trp1-Cre+/tg; Tsc1flox/flox mice suffer from deregulated mTORC1 activity in a subset of peripheral, ciliary margin-derived RPCs and this results in "clonal expansion" of these mutant cells into the central wild-type retina. This phenotype is suggested to cause retinal laminar disorganization in the form of rosette structures.

2. Upon terminal differentiation of these rapidly diving CKO cells, MG are produced in excess, but eventually undergo apoptosis.

3. MG apoptosis is due to "mitotic aging" and resulting senescence-associated cell death.

4. These phenotypes are specific to the RPCs of the Trp1-Cre line and are not produced by other RPC Cre lines (Chx10-Cre, aPax6-Cre, and mRx-Cre).

5. Tsc1 CKO RPC hyperproliferation is dependent on HIF1a activity, which activates glycolysis required to drive that cellular state.

While authors present a large body of work, and they are clearly focused on identifying the precise molecular mechanism at play, it is still not clear to this reviewer what are primary versus secondary aspects of this phenotype. Lingering questions regarding the specificity of Trp1-Cre leave some doubt as to the accuracy of the interpretations provided in what is otherwise an experimentally sound paper.

The main, unresolved conundrum is the lack of RPC phenotype when using the Chx10-Cre, mRx-Cre, and aPax6-Cre lines. I understand the authors' argument that since Trp1-Cre is active in fewer RPCs, the CKOs cells may be allowed to expand into the territory of the more slowly dividing, centrally located WT cells. However, if this is indeed the case, why was it not observed in the Chx10-Cre or aPax6-Cre mice? My lab and others have used these line for years and we can state with absolute certainty that the Cres are mosaically expressed. Therefore, one would definitely end up with patches of CKOs cells adjacent to WT cells. This is particularly true for aPax6-Cre that is predominantly expressed in peripheral RPCs. Wouldn't the CKO patches overtake the WT patches centrally? The most parsimonious explanation for the discrepancies between the Trp1-Cre and the other RPC Cre lines, is that something other than Tsc1 CKO is contributing to the phenotype. I have several concerns regarding the Trp1-Cre line. Thanos, et al. (2012) reports that the line exhibits Cre toxicity leading to "RPE dysfunction and concomitant disorganization of RPE layer morphology, large areas of RPE atrophy, retinal photoreceptor dysfunction, and microglial cell activation in the affected areas". While the authors seem to be addressing this issue by comparing Trp1-Cre; Tsc1+/flox mice to Trp1-Cre; Tsc1flox/flox mice, are they certain that the mice in question all carry one copy of the Trp1-Cre transgene? This line is a random transgene insertion. Therefore, if one does not know where the transgene inserted into the genome, it's impossible to genotype for 1 versus 2 copies and a test cross would have to be used to determine zygosity. Different dosages of Cre could impact toxicity effects. Also, if the transgene inserted into an essential gene that could also lead to a phenotype in a homozygous state. The authors should also consider the genetic background of these mice as several rd alleles are known to be present within certain inbred strains (i.e. rd8 in C57BL/6N).

This reviewer recognizes the tremendous effort put forth in this paper and sincerely appreciates it. The utilization of mouse genetics is truly impressive, and I commend you for that. Nevertheless, I still cannot shake the concern that there is something about the Trp1-Cre line that is misleading you. It's very odd to me that the other RPC Cre lines show absolutely no phenotype in vivo. I understand that this is not an easy question to answer, but it would benefit your paper greatly if you could show some direct evidence that the Trp1-Cre mediated phenotype is specific to Tcs1 loss in RPCs without any contribution from negative effects of the Trp1-Cre itself. It's formally possible that Trp1-Cre on its own has an effect that's synergizing with Tsc1 loss to give you the phenotype.

You could employ the Trp1-CreERT2 line (Mori, et al., 2012). This an independent transgene and the random integration site is expected to be completely different from the Trp1-Cre transgene. Also, because Cre activity is tamoxifen-dependent, this line would be an effective means to confirm the Trp1-Cre Tsc1 CKO phenotype as being specific to Trp1-Cre mediated recombination of the floxed allele. In other words, the comparison would be Trp1-CreERT2; Tsc1flox/flox littermates with or without tamoxifen. However, there could still be issues with Cre toxicity upon induction. Another method could be AAV-Cre which could be used to sparsely infect RPCs with Cre followed by assessment of CKO cell expansion into WT areas. This is probably the preferred method as it does not depend on the Trp1 promoter.

Figure S1.

Here the authors show pS6 immunofluorescence to indicate expression in the retina but not the ciliary margin or RPE. Why is the retina staining so sparse? Wouldn't one expect more widespread expression with the proliferative RPC population? What is the identity of the pS6+ cells?

Figure 1.

Is the ciliary body actually "absent" as the authors state or is it malformed?

At P14 and P30, if MGs are missing due to apoptosis, why are other retinal cell types that require MGs for homeostatic support also not degenerative?

Might the MG loss actually just be a downregulation of *Sox9* and GS expression during gliosis? GFAP, Cyclin D1, and p27 immunofluorescence could address this. However, it's also curious that in Figure S2 the authors don't show an increase in GFAP expression in the MG of the CKO mice. Such disruptions in retinal architecture would certainly be expected to result in a gliotic response of whatever MGs are still present near the affected area.

Figure 3.

The *Sox9*+ MG layer in the Chx10-Cre; Tsc1flox/flox mice looks very disorganized. Is this a representative image? If it is, it might reflect a subtle phenotype. The MG layer sometimes appears less organized when undergoing a gliotic response.

Figure 4.

Please indicate statistical significance in panels E, G, and H.

I take issue with the term "clonal expansion" as that's not really what this experiment examines. A replication incompetent retroviral lineage trace would directly examine clonal expansion. Furthermore, it would allow the authors to determine how many times a particular RPC divides. This relates to this idea of the mutant RPCs as being "mitotically old". How many divisions does it take for an RPC to reach this point? Why would reaching a mitotic limit and then producing a post-mitotic MG result in a degenerative MG? Might the death of the MGs that is observed at P14 and P30 simply be a pruning mechanism to rid the retina of excessive MGs that are not needed?

Figure 5.

Why didn't the authors do the Tsc1/Raptor double CKO and assess for rescue? This seems to be the most direct approach and implicate mTORC1 loss in the primary phenotype. The Hif1a CKO rescue may be indirect.

The Psmb-TKO rescue of the Tsc1 CKO phenotype is curious. Why would only a quarter rescue? Might the extensive crosses to generate this mouse have resulted in the segregation of a deleterious allele linked to the Trp1-Cre or different mice with 1 or 2 copies of Trp1-Cre?

Figure S6.

There is no indication of statistical significance.

Figure S11.

Glycolysis is the predominant mode of ATP synthesis in proliferative neural progenitor cells. Therefore, it stands to reason that lactate production was increased in the Tcs1 CKOs which the authors claim has accelerated cell cycle progression. However, it's not clear whether this is a primary or secondary phenotype.

*Reviewer #3:*

This report is a follow-up from an earlier study, in which the same lab reported Tsc1 cko leads to accelerated cell cycle progression. In this study, they repeat some of the same experiments, but they now show the partial rescue of the accelerated cell cycle progression with the concomitant cko of Hif1a (Figure 6), the Muller glial degeneration and retinal rosetting as the animals mature (Figure 1) and the dependence of the phenotype on the different cre-lines (Figure 4). These are all interesting phenotypes/results, but there are also some concerns, listed below.

1. The evidence for selective Muller glial cell death is not that strong. The demonstration of active Caspase in Muller glia in the supplement was not clear. Additional/better examples are needed. In addition, while rosettes can result from the loss of Muller glia, they can also result from increased or prolonged Muller glial proliferation (eg. p27 ko) or from inhibition of the BMP pathway during development, among other things. They use only *Sox9* to label the Muller glia and the disruption in normal retinal histology might make accurate quantitation difficult. Additional markers and better characterization of the glial cells prior to their overt loss (earlier time points) would help understand the phenotype.

2. The evidence that the developmental over-proliferation of the Tsc1 cko progenitors and the Muller glial degeneration are linked is partly supported by the fact that deletion of Tsc1 in mature Muller glia did not cause this phenotype. However, it is not clear that the Tsc1 was effectively deleted in this experiment. More evidence to show this experiment actually produced the deletion should be provided.

3. One of the most interesting aspects of the report is that only the deletion of Tsc1 using the Trp1-cre line leads to the phenotypes. One interpretation of this is that the hyper-proliferation of the Tsc1 cko progenitors only occurs when there are normal cells nearby. The α-Pax6cre line shows a reduced phenotype, consistent with this idea, while when Tsc1 is knocked out in all progenitors across the retina, proliferation appears to be normal. They interpret these different phenotypes in terms of the "age" of the progenitors or the number of total cell divisions, but it seems more consistent with a model where normal and Tsc1 deficient progenitors compete for space and the Tsc1 deficient cells outcompete normal cells, but not other Tsc1 deficient progenitors. This alternative model could be tested by looking more closely at the percentages of EdU+ cells near the boundary of the conditional deletion. Probably best to do this early in the α-Pax6cre line where the boundary is likely to be sharp.

4. The neurosphere assay is not very quantitative to show differences between the Tsc1 cko (Figure 4), particularly when they do not see differences until the spheres are grown for 4 weeks, long past when proliferation would normally have finished in the retina. The in vivo cell cycle analyses in Figure 7 do a better job anyway.

5. They rescue the rosetting phenotype by crossing the Tsc1 cko mice with a Hif1a deletion. The "hyper-expansion" phenotype is less well rescued (Figures 6B, 6D). Thus it seems like these two phenotypes might be un-related. The authors should discuss this possibility.

6. The authors argue that Muller glia undergo senescence because they are the last cell type generated and the progenitors have exceeded their mitotic limit.This is not exactly true. Muller glia are among the last cells generated, but rods and bipolar cells are also included in the last cell divisions. However, these other last generated cells do not seem to be undergoing cell death or senescence-related gene expression. This is specifically demonstrated in supplemental Figure S9. Rather it appears that there is a specific requirement for Tsc1 in Muller glia, perhaps near the end of neurogenesis, much like there is for p27kip. It would be interesting to determine whether the premature end of neurogenesis might lead to incomplete differentiation of the Muller glia.

[Editors' note: further revisions were suggested prior to acceptance, as described below.]

Thank you for submitting your article "mTORC1-induced retinal progenitor cell overproliferation leads to accelerated mitotic aging and degeneration of descendent Müller glia" for consideration by *eLife*. Your article has been reviewed by 3 peer reviewers, and the evaluation has been overseen by a Reviewing Editor and Carlos Isales as the Senior Editor. The reviewers have opted to remain anonymous.

Essential revisions:

1) Please discuss the potential reasons as of why Chx10-Cre null neurospheres divide better in vitro as indicated by reviewer 1.

2) Please discuss why Tcs1 CKOs MG (based on tdTomato Cre reporter expression) seem to be unable of undergoing reactive gliosis, as pointed out by reviewer 2.

3) Please change the statement on page 21 that "The MG are the last-born retinal cell type, arising around the first postnatal week in mice." As reviewer 3 indicate this is not correct. Also discuss why rod and bipolar that are generated around the same time as Müller glia do not undergo the same apoptosis.

4) Please also discuss possible peculiarities of the ciliary margin, as indicated by reviewer 3.

*Reviewer #1:*

The authors have added important new experiments. First, they showed that using a TRP1-Cre-ER system, in which they activate Cre at E9.5 with tamoxifen, they obtain similar results to that observed with the TRP1-Cre system. This result addresses the concerns of the other reviewers that the integration site of TRP1-Cre may have been contributing to the phenotype.

Second, they isolated E13 progenitors from mice in which Tsc1 was deleted using Chx10-Cre, which does not induce the over-proliferation/Muller glia degeneration phenotype in vivo, and assessed resultant neurosphere size and number. They compared Tsc1 null to Tsc1 heterozygous neurospheres. Interestingly, Tsc1 null progenitors generated ~2-fold more neurospheres, and the spheres were somewhat (~20%) larger than those from Tsc1 hets. They then degenerated, mimicking the effect seen in vivo when TRP1-Cre is used to knockout Tsc1. These data imply that the effects of the knockout are cell autonomous, and are not a consequence of cell competition.

I commend the authors for their extra work to address the concerns of the reviewers, and the paper should be published. However, I do feel there should be an addition to the Discussion. The neurosphere/Chx10-Cre result also suggests that there is something about the in vivo milieu of the mice lacking the proliferation phenotype (Chx10-Cre, Rx-Cre etc) that suppresses the proliferative effect of Tsc1 deletion. One possibility, raised in the author's rebuttal, is that the ratio of mutant to normal cells must be low for the proliferative defect to occur, implying either that normal cells stimulate the mutant cell division, or excess mutant cells inhibit over-proliferation. Against the idea that normal cells stimulate over-proliferation, they now show that Chx10-Cre null neurospheres divide better in vitro. That leaves the notion that high numbers of Tsc1 cells may inhibit their own proliferation. The Discussion is completely lacking any mention of this confusing and unsolved aspect of the data, nor does it even acknowledge the paradox. At the very least the authors could highlight the problem and state that the difference in the effects of various Cre models in vivo coupled with the neurosphere data in vitro is puzzling, and the mechanism as to why mutant progenitors must be present at a low frequency to over-proliferate is unresolved.

*Reviewer #2:*

The reviewers have satisfied most of my concerns.

However, the mutant GFAP immunofluorescence raises an interesting question. The authors clearly show the presence of residual *SOX9*+ MGs within the mutant retinae with lamination defects. However, these mutant retinae, which are clearly experiencing damage, do not upregulate GFAP. These data suggest that these presumably Tcs1 CKOs MG (based on tdTomato Cre reporter expression) are not capable of undergoing reactive gliosis. Is MG differentiation, or the ability to detect damage, or Gfap transcription compromised?

*Reviewer #3:*

The authors have done a very good job at addressing my previous concerns. However, I think the text needs a few changes for accuracy. The authors state on page 21 that "The Mg are the last-born retinal cell type, arising around the first postnatal week in mice." This is not correct. The MG are AMONG the last cells born in the retina, but rods and bipolar cells are included in two cell clones that contain Muller glia. Therefore, the authors should change this sentence accordingly. Moreover, since at least some of the rods and most of the bipolar cells have been generated in the KO mice by progenitors that have undergone as many rounds of division as those that generated the MG, one might presume their survival would be similarly affected. Yet they appear not to undergo the apoptosis that the MG are subject to in these mice. The authors should discuss this point in the Discussion.

The second point that could bear some elaboration is the possibility that there is something unique about the Trp-cre expressing cells at the retinal margin. The authors show in Figure 4 that there is a progressive decline in RPC overgrowth from peripheral to central retina when the cre is targeted to progressively more RPCs, but these different cre lines also have a relative decrease in the contribution of RPCs adjacent to the CM. It is possible that this phenotype is primarily due to expansion in the proliferation of a relatively small subset of RPCs, ones that may more closely resemble the CMZ cells of lower vertebrates. The authors should consider/discuss this possibility.

---

## [Author Response]

Essential revisions:There is general agreement that the authors report a fascinating phenotype. However, there are also several concerns that are listed in the individual reviewers' reports. The common and most pressing unresolved issues that the authors should experimentally address are the following:

We appreciate the editors and reviewers for evaluating our report positively. We also thank for the comments that could certainly improve the quality of our paper.

1) Is the Trp1-Cre line somehow contributing to the phenotype? This could be solved by the use of theTrp1-CreERT2 line or the injection so AAV-Cre. Extensive discussion of this issue is provided in the individual reviewers' assessment of the manuscript.

We have also been aware of the toxicity issue of TRP1-Cre. Thus, we compared the phenotypes of Tsc1(f/+);TRP1-Cre and Tsc1(f/f);TRP1-Cre littermate mice to exclude the possibility that TRP1-Cre causes the phenotypes. In addition, those mice were obtained from the breeding of Tsc1(f/f) and Tsc1(f/+);TRP1-Cre pairs. Therefore, all mice carrying TRP1-Cre transgene are heterozygous for TRP1-Cre, and this could have reduced toxicity of TRP1-Cre.

As the reviewer recommended, we analyzed the phenotypes of Tsc1(f/f);TRP1-CreERT2 mice, in which Tsc1 was deleted in the CM and RPE populations in tamoxifen-dependent manner. We found that deletion of Tsc1 in early embryonic stages by E9.5 results in MG degeneration and retinal rosettes in R26tdTom-positive clones in the mature retina whereas the deletion in later stages did not (Figure 4 —figure supplement 2). The results suggest that Tsc1 deletion in early CM progenitors is necessary to provide a chance for their descendent RPCs to divide exceeding the division limit.

2) Can the results be interpreted in terms of cell competition? The study does not provide sufficient support to the statement that telomere shortening is a "cell autonomous" effect.

We agree to the reviewers’ opinions on the conclusion that the fast-dividing Tsc1-deficient clones might win the competition with slowly-dividing WT clones for occupying a space in the retina. This might result in the over proliferation of Tsc1-deficient RPCs exceeding the division limit. Supporting this, telomere shortening occurred excessively in Tsc1-deficient retinal cells in comparison to the telomeres of their neighboring WT cells (Figure 4B). However, we could not find evidence that compensatory hyper expansion and telomere shortening occur in WT cells in Raptor(f/f);TRP1-Cre mouse retina, in which Raptor-deficient cells were depleted precociously (Figure 5H). These results suggest that the overproliferation, which leads to excessive telomere shortening, is an autonomous phenomenon of Tsc1-deficent cells but a general feature of RPCs, which are surrounded by slowly-dividing neighboring RPCs.

In the revision, we also co-cultured Tsc1-deficient RPCs and WT RPCs to form neurospheres in neighbor, respecting the reviewer’s suggestion. We found the neurospheres of Tsc1-deficient RPCs expand faster than WT neurospheres, which present in neighbor or in a separate space (Figure 4 —figure supplement 3A – 3C). We also found the neurospheres derived from Tsc1-deficient RPCs exhibit the characteristics of mitotic aging, including telomere shortening, senescence, and apoptosis, with or without WT neurospheres (Figure 4 —figure supplement 3D – 3F). Therefore, the results also suggest that the clonal expansion and mitotic aging are intrinsic properties of Tsc1-deficient RPCs, which just need a space to expand their clones in the retina.

Reviewer #1:[…]– It is interesting, but overlooked in the paper, that the telomere length in KO cells is not intrinsically shorter in KO cells (e.g. not in mRx-Cre KO or CHx10-Cre KO retina), but only in retinas with mixed wt and KO cells (significant in TRP1-Cre KO, and n.s. in α-Cre but mean is lower). That finding suggests that the telomere shortening may be dependent on the presence of neighboring WT cells. This could, for example, reflect "cell competition" between KO and WT cells, in which the elevated rates of division is due to greater fitness of KO cells, that is stimulated by their weaker neighbors.

We agree to the reviewer’s opinion on the conclusion of competition between KO and WT cells for a space in the retina. As we wrote in the text, the first paragraph of section starts by “MG degeneration is correlated… (page 9)”, the ratio of fast-dividing Tsc1-deficient RPCs to slowly-dividing WT RPCs is likely a critical factor that determines whether the Tsc1-deficient RPCs can over proliferate and expand their clones exceeding the division limit. Thus, the Tsc1-deficient RPCs should be present in minority in the retina to have a competitive position in clonal expansion in the limited space of the retina, as those are in Tsc1(f/f);TRP1-Cre mice. On the contrary, in the other Tsc1-cko mice, the neighboring RPCs of a Tsc1-deficient RPC are mostly Tsc1-deficient, too, thus the RPC cannot win the competition with its neighbor. This difference might allow Tsc1-deficient RPCs to over proliferate and exceed the division limit only in Tsc1(f/f);TRP1-Cre mouse retina.

Indeed, mTOR signaling has been linked to cell competition in mammals: PMID: 29720666; PMID: 33652024. It should be feasible to assess this notion by isolating early stage KO cells and assessing whether their division rate and/or telomere attrition is boosted specifically in the presence of WT cells relative to pure KO cultures.

Respecting the reviewer’s suggestion, we co-cultured Tsc1-deficient RPCs and WT RPCs to form neurospheres in neighbor. We found the neurospheres of Tsc1-deficient RPCs expand faster than the neighboring WT neurospheres (Figure 4 —figure supplement 3A – 3C). We also found the neurospheres derived from Tsc1-deficient RPCs exhibit the characteristics of mitotic aging, including telomere shortening, senescence, and apoptosis, with or without WT neurospheres (Figure 4 —figure supplement 3D – 3F). Therefore, the results also suggest that the clonal expansion and mitotic aging are intrinsic properties of Tsc1-deficient RPCs. This idea was also supported by the finding that WT RPCs did not over proliferate to show extra telomere shortening in Raptor-cko mouse retina, where Raptor-deficient RPCs are depleted precociously.

– After showing telomere shortening in the KO but not wt cells from the same TRP1-Cre retina, the papers states that this phenomenon must be "cell autonomous". If that is the case, why is there no telomere shortening in other strains in which the entire/most of the retia is KO? As noted above, it appears that the shortening may depend on WT neighbors, in which case the term "cell autonomous" would be inappropriate.

We think the space where an RPC can expand is likely a limiting factor for clonal expansion. For a Tsc1-deficient RPC in Tsc1(f/f);TRP1-Cre mouse retinas, majority neighboring RPCs are slowly-expanding WT RPCs, thus they could have a competitively dominant position to expand fast in a retinal space. However, this chance for a Tsc1-deficient RPC should be much lower in the other Tsc1-cko mouse retinas, since its neighbors are also mostly fast dividing Tsc1-deficient RPCs. Thus, the Tsc1-deficient RPC cannot expand faster than its neighbors in other Tsc1-cko mouse retinas.

– The same is true of other biochemical phenotypes that are unique to the TRP1-Cre retina, notably senescence. It's feasible, therefore, that the entire basis for the phenotypic specificity in the TRP1-Cre retina is dependent on mutant/normal cell interaction. The authors should attempt assays to validate or rule out that possibility. In addition to the cell mixing studies above, it should also be feasible to perform clonal studies in explants or in vivo using retroviral Cre delivery. It would, for example, be feasible to at least assess whether the resulting clones contain senescent cells (e.g. SA-betagal-positive staining).

Owing to technical limitations, we could not deliver the virus into the eyes of early mouse embryos in the uterus. Instead, following the reviewer #2’s suggestion, we employed TRP1-CreERT2 mice to delete Tsc1 sparsely in the retina. We could find the degeneration of MG and rosette formation in the CreER-affected areas in Tsc1(f/f);TRP1-CreER mouse retinas (Figure 4 —figure supplement 2).

– The authors show that expressing Tert does not rescue elevated cell death in the TRP1-Cre KO retina. They conclude on page 12: "This suggests that telomere shortening is unlikely to be the cause of MG degeneration in Tsc1fl/fl;TRP1-Cre mice". They would need to demonstrate that telomere length is rescued to make that claim, otherwise they should modify to state: "This may mean either that Tert expression is insufficient to rescue telomere length in this context, or that telomere shortening does not cause MG degeneration"

We measured the telomere length of P30 dTert-TG and Tsc1-cko;dTert-TG mouse retinas, and found they have longer telomeres than WT and Tsc1-cko mice, respectively (Figure 5 —figure supplement 2C). The results suggest that telomere shortening is unlikely a direct cause of the phenotypes. We also modified the expression as “This suggests that telomere shortening is not a direct cause of MG degeneration in Tsc1fl/fl;TRP1-Cre mice”.

Overall, the paper is really interesting, but the authors have overlooked an explanation for their results that should be addressed, given the prominent literature on cell competition, and its link to mTOR.Reviewer #2:[…]The main, unresolved conundrum is the lack of RPC phenotype when using the Chx10-Cre, mRx-Cre, and aPax6-Cre lines. I understand the authors' argument that since Trp1-Cre is active in fewer RPCs, the CKOs cells may be allowed to expand into the territory of the more slowly dividing, centrally located WT cells. However, if this is indeed the case, why was it not observed in the Chx10-Cre or aPax6-Cre mice? My lab and others have used these line for years and we can state with absolute certainty that the Cres are mosaically expressed. Therefore, one would definitely end up with patches of CKOs cells adjacent to WT cells. This is particularly true for aPax6-Cre that is predominantly expressed in peripheral RPCs. Wouldn't the CKO patches overtake the WT patches centrally?

All Tsc1-cko mice, which have the mosaicism of Cre activity, showed the expansion of Tsc1-deficient clones in their retinas (Figure 4, A – C), however only the deletion by TRP1-Cre resulted in the MG degeneration and retinal rosettes. We think the chance for a Tsc1-deficient RPC to expand its clone is dependent of the expansion power of neighboring RPCs. For a Tsc1-deficient RPC in Tsc1(f/f);TRP1-Cre mouse retinas, majority neighboring RPCs are slowly expanding WT RPCs, thus it could have a strong competitive position to expand among those WT RPCs. The WT cells unaffected by aPax6-Cre and Chx10-Cre are, however, less than 40% and 20%, respectively, thus majority neighbors of a Tsc1-deficient RPC are also Tsc1-deficient RPCs. Thus, the Tsc1-deficient RPC could not win a competition with neighboring Tsc1-deficent RPCs to expand in a retinal space.

The most parsimonious explanation for the discrepancies between the Trp1-Cre and the other RPC Cre lines, is that something other than Tsc1 CKO is contributing to the phenotype. I have several concerns regarding the Trp1-Cre line. Thanos, et al. (2012) reports that the line exhibits Cre toxicity leading to "RPE dysfunction and concomitant disorganization of RPE layer morphology, large areas of RPE atrophy, retinal photoreceptor dysfunction, and microglial cell activation in the affected areas". While the authors seem to be addressing this issue by comparing Trp1-Cre; Tsc1+/flox mice to Trp1-Cre; Tsc1flox/flox mice, are they certain that the mice in question all carry one copy of the Trp1-Cre transgene? This line is a random transgene insertion. Therefore, if one does not know where the transgene inserted into the genome, it's impossible to genotype for 1 versus 2 copies and a test cross would have to be used to determine zygosity. Different dosages of Cre could impact toxicity effects. Also, if the transgene inserted into an essential gene that could also lead to a phenotype in a homozygous state.

We have also been aware of the toxicity issue of TRP1-Cre. Thus, we compared the phenotypes of Tsc1(f/+);TRP1-Cre and Tsc1(f/f);TRP1-Cre littermate mice to exclude the possibility that TRP1-Cre causes the phenotypes. In addition, those mice were obtained from the breeding of Tsc1(f/f) and Tsc1(f/+);TRP1-Cre pairs. Therefore, all mice carrying TRP1-Cre transgene are heterozygous for TRP1-Cre, and this could have reduced toxicity of TRP1-Cre.

The authors should also consider the genetic background of these mice as several rd alleles are known to be present within certain inbred strains (i.e. rd8 in C57BL/6N).

We believe the genetic background of the mice converged to C57BL/6J after repeated backcrossing (>10 generation). We have also confirmed the mice used for the study do not have rd mutations, such as rd1, rd10, and rd8 (data not shown), by genotyping.

This reviewer recognizes the tremendous effort put forth in this paper and sincerely appreciates it. The utilization of mouse genetics is truly impressive, and I commend you for that. Nevertheless, I still cannot shake the concern that there is something about the Trp1-Cre line that is misleading you. It's very odd to me that the other RPC Cre lines show absolutely no phenotype in vivo. I understand that this is not an easy question to answer, but it would benefit your paper greatly if you could show some direct evidence that the Trp1-Cre mediated phenotype is specific to Tcs1 loss in RPCs without any contribution from negative effects of the Trp1-Cre itself. It's formally possible that Trp1-Cre on its own has an effect that's synergizing with Tsc1 loss to give you the phenotype.

We have also been aware of the toxicity issue of TRP1-Cre. Thus, we compared the phenotypes of Tsc1(f/+);TRP1-Cre and Tsc1(f/f);TRP1-Cre littermate mice to exclude the possibility that TRP1-Cre causes the phenotypes. We have also confirmed the mice used for the study do not have rd mutations, such as rd1, rd10, and rd8 (data not shown), by genotyping.

You could employ the Trp1-CreERT2 line (Mori, et al., 2012). This an independent transgene and the random integration site is expected to be completely different from the Trp1-Cre transgene. Also, because Cre activity is tamoxifen-dependent, this line would be an effective means to confirm the Trp1-Cre Tsc1 CKO phenotype as being specific to Trp1-Cre mediated recombination of the floxed allele. In other words, the comparison would be Trp1-CreERT2; Tsc1flox/flox littermates with or without tamoxifen. However, there could still be issues with Cre toxicity upon induction. Another method could be AAV-Cre which could be used to sparsely infect RPCs with Cre followed by assessment of CKO cell expansion into WT areas. This is probably the preferred method as it does not depend on the Trp1 promoter.

As the reviewer recommended, we analyzed the phenotypes of Tsc1(f/f);TRP1-CreERT2 mice, in which Tsc1 was deleted in the CM and RPE populations in tamoxifen-dependent manner. We found that deletion of Tsc1 in early embryonic stages by E9.5 results in MG degeneration and retinal rosettes in R26tdTom-positive clones in the mature retina whereas the deletion in later stages did not (Figure 4 —figure supplement 2). The results suggest that Tsc1 deletion in early CM progenitors is necessary to provide a chance for their descendent RPCs to divide exceeding the division limit. However, owing to technical limitations, we could not deliver AAV-Cre virus into the eyes of early mouse embryos across the uterus.

Figure S1.Here the authors show pS6 immunofluorescence to indicate expression in the retina but not the ciliary margin or RPE. Why is the retina staining so sparse? Wouldn't one expect more widespread expression with the proliferative RPC population? What is the identity of the pS6+ cells?

Previous reports have also shown the sparse pS6 signals in developing mouse retina (Choi et al., 2018; Hagglund et al., 2017). The pS6 signals were detected in RPCs (Vsx2-positive) and PMNs (Tuj1-positive) (Figure 1 —figure supplement 1). The former pS6 signals might be related with the acceleration of cell cycle, while the latter might be related with the growth and maturation of PMNs. The mTORC1 activity can be changed dynamically during cell cycle, thus pS6 might be detectable at high in sub-RPC population at certain cell cycle.

Figure 1.Is the ciliary body actually "absent" as the authors state or is it malformed?

We examined the distribution of CB markers, such as Aqp1 and Otx1, in the mouse eye sections. The Aqp1- and Otx1-positive cells were detectable in reduced numbers in the peripheral parts of Tsc1(f/f);TRP1-Cre mouse eyes, suggesting the hypoplasia of CB (please see Author response image 1). We, thus, modified the expression that “the ciliary body is malformed”, in the revised text.

**Author response image 1. sa2fig1:** CB malformation in Tsc1^f/f^;TRP1-Cre mouse eyes. (A) Distribution of CB cells in the littermate mouse eyes was investigated by immunostaining of specific markers, such as Otx1 and Aqp1. (B) Numbers of Otx1-stained nuclei in the peripheral part of the eye sections were counted and relative numbers against Tsc1^f/+^;TRP1-Cre samples are shown in the graph. **, p<0.01; ***, p<0.001.

At P14 and P30, if MGs are missing due to apoptosis, why are other retinal cell types that require MGs for homeostatic support also not degenerative?

Other retinal cell types also degenerated later in the Tsc1(f/f);TRP1-Cre mouse retinas (please see Author response image 2). However, those retinal cells were not affected by P14, when MG loss was already evident (Figure 1D; Figure1 —figure supplement 2A). The results suggest that the degeneration of retinal neurons might start when the structural and functional changes caused by MG loss are accumulated in the retina.

**Author response image 2. sa2fig2:** Degeneration of retinal neurons in Tsc1^f/f^;TRP1-Cre adult mice. (A) Distribution of Cre-affected cells in 9 months-old littermate mouse retinas are visualized by R26tdTom reporter. Activation of mTORC1 in the retinas were determined by pS6 immunostaining. (B) Distribution of retinal cell type-specific markers (explained in Figure 1 and Figure 1 —figure supplement 2) in the retinas were also investigated by immunostaining.

Might the MG loss actually just be a downregulation of Sox9 and GS expression during gliosis? GFAP, Cyclin D1, and p27 immunofluorescence could address this. However, it's also curious that in Figure S2 the authors don't show an increase in GFAP expression in the MG of the CKO mice. Such disruptions in retinal architecture would certainly be expected to result in a gliotic response of whatever MGs are still present near the affected area.

We provide the immunostaining results of other MG markers, such as p27 and *Sox2*, in the revised Figure 1 —figure supplement 2A. The quantifications of the results are also provided in Figure 1D.

We also co-stained Gfap and *Sox9* to determine the gliotic responses of MG in mouse retina at P30, when Gfap-positive signals are observed in Tsc1(f/f);TRP1-Cre mouse inner retina (Figure 1 —figure supplement 2A). However, those inner retinal Gfap signals are mostly *Sox9*-negative (please see author response image 3), suggesting that those are not MG but might be the extension of astrocyte cell processes.

**Author response image 3. sa2fig3:** Activation and gliosis of MG in the mouse retinas were determined by co-staining of *Sox9*, a MG marker, and Gfap, which is expressed in astrocytes and activated MG.

Figure 3.The Sox9+ MG layer in the Chx10-Cre; Tsc1flox/flox mice looks very disorganized. Is this a representative image? If it is, it might reflect a subtle phenotype. The MG layer sometimes appears less organized when undergoing a gliotic response.

Nuclear positions of the INL cells are disorganized in Tsc1-cko mouse retinas because of the cytomegaly and excessive branching of Tsc1-deficient retinal neurons and MG (Figure 1C; Figure1 —figure supplement 2A). Similar results have also been reported previously (Choi et al. (2018); Hagglund et al. (2017)).

Figure 4.Please indicate statistical significance in panels E, G, and H.

We added the p-values in the graphs.

I take issue with the term "clonal expansion" as that's not really what this experiment examines. A replication incompetent retroviral lineage trace would directly examine clonal expansion. Furthermore, it would allow the authors to determine how many times a particular RPC divides.

It is necessary to inject the virus at low copy into the early mouse embryonic mouse eyes across the uterus. Owing to a technical limitation, we could not deliver a replication incompetent retrovirus into early mouse embryonic eyes. We, instead, used TRP1-CreERT2, to address this point (please see our response to your major comment above).

This relates to this idea of the mutant RPCs as being "mitotically old". How many divisions does it take for an RPC to reach this point?

Unfortunately, we do not know the exact number of mouse RPC division limit, which should be counted from one cell stage of the embryo. Instead, we assessed relative division numbers by calculating the cells comprising a neurosphere that reaches the maximum number (Figure 4F – 4G; Figure 4 —figure supplement 3).

Why would reaching a mitotic limit and then producing a post-mitotic MG result in a degenerative MG? Might the death of the MGs that is observed at P14 and P30 simply be a pruning mechanism to rid the retina of excessive MGs that are not needed?

Excessive retinal cells produced during development are known to degenerate in the postnatal stages as the reviewer indicates. This developmental pruning leaves the cells in constant numbers at the end. The numbers of MG were much less than the normal in the mature Tsc1(f/f);TRP1-Cre mouse retinas in comparison to their littermates’ (i.e., P14 and P30), whereas they were more in the developing retina (i.e., P7) (Figure 1D; Figure 1 —figure supplement 2). The numbers of other retinal cell types were, however, not changed significantly, although they were also produced at higher numbers ahead of their regular schedules (Choi et al., 2018; Figure 1D; Figure 1 —figure supplement 2). Therefore, we think the loss of MG in the Tsc1(f/f);TRP1-Cre mouse retina is unlikely to result from excessive pruning.

Figure 5.Why didn't the authors do the Tsc1/Raptor double CKO and assess for rescue? This seems to be the most direct approach and implicate mTORC1 loss in the primary phenotype. The Hif1a CKO rescue may be indirect.

The phenotypes of Tsc1/Raptor double cko mouse retinas were not greatly different from those of Raptor-cko mouse retinas. We provide the results in Author response image 4.

**Author response image 4. sa2fig4:** Eye and retinal structures of P30 littermate mice with indicated genotypes were investigated by H&E staining. Distribution of the Cre-affected cells and mTORC1 activation of the cells were examined by the immunostaining of R26EYFP and pS6, respectively.

The Psmb-TKO rescue of the Tsc1 CKO phenotype is curious. Why would only a quarter rescue?

We think the incomplete rescue might be related with the adaptation of RPCs, which might utilize alternative proteasome machineries in the absence of the immunoproteasome and over proliferate. The alternative proteasomes were, however, not able to compensate the immunoproteasome in a quarter, thus the phenotypes cannot appear in their retinas.

Might the extensive crosses to generate this mouse have resulted in the segregation of a deleterious allele linked to the Trp1-Cre or different mice with 1 or 2 copies of Trp1-Cre?

The mice were obtained from the breeding of Tsc1(f/f) and Tsc1(f/+);TRP1-Cre pairs. Therefore, all mice carrying TRP1-Cre allele are TRP1-Cre(+/-).

Figure S6.There is no indication of statistical significance.

We added the p-values in the graph.

Figure S11.Glycolysis is the predominant mode of ATP synthesis in proliferative neural progenitor cells. Therefore, it stands to reason that lactate production was increased in the Tcs1 CKOs which the authors claim has accelerated cell cycle progression. However, it's not clear whether this is a primary or secondary phenotype.

We think the increase of lactate production is a secondary phenotype caused by the shortage of intracellular ATP, which was resulted from enhanced energy consumption of Tsc1-deficient RPCs.

Reviewer #3:This report is a follow-up from an earlier study, in which the same lab reported Tsc1 cko leads to accelerated cell cycle progression. In this study, they repeat some of the same experiments, but they now show the partial rescue of the accelerated cell cycle progression with the concomitant cko of Hif1a (Figure 6), the Muller glial degeneration and retinal rosetting as the animals mature (Figure 1) and the dependence of the phenotype on the different cre-lines (Figure 4). These are all interesting phenotypes/results, but there are also some concerns, listed below.1. The evidence for selective Muller glial cell death is not that strong. The demonstration of active Caspase in Muller glia in the supplement was not clear. Additional/better examples are needed.

We determined the apoptotic cell death of MG by TUNEL assay and immunostaining of active caspase-3 (Casp-3), which are generally used for the detection of apoptotic cells. In the original Figure S3, the apoptotic MG are, however, not identified clearly by Casp-3 immunostaining. Thus, we modified the protocol for the staining to visualize Casp-3 more clearly (please see revised Figure 1 —figure supplement 3A). In the modified staining condition, *Sox9* was, however, not detectable, thus *Sox2* was used to identify MG in the sections.

In addition, while rosettes can result from the loss of Muller glia, they can also result from increased or prolonged Muller glial proliferation (eg. p27 ko) or from inhibition of the BMP pathway during development, among other things.

We could not find EdU incorporation in GS-positive MG cells in P14 mouse retinas (please see Author response image 5), suggesting that there was no prolonged MG proliferation in the mouse retinas.

**Author response image 5. sa2fig5:** Proliferation of MG in P14 littermate mouse retinas was determined by the presence of GS-positive MG, which incorporated EdU for 24h.

We could not find the changes of phosphorylated Smad1 and 5 (pSmad1/5), which reflect the activation of BMP signaling, in the mouse retinas (please see Author response image 6).

**Author response image 6. sa2fig6:** Activity of BMP signaling pathway in mouse retinas were determined indirectly by detecting phosphorylated Smad 1 and 5 (pSmad1/5), which are induced by BMP-activated receptors.

They use only Sox9 to label the Muller glia and the disruption in normal retinal histology might make accurate quantitation difficult. Additional markers and better characterization of the glial cells prior to their overt loss (earlier time points) would help understand the phenotype.

We have also detected MG by additional markers, including GS, p27, and *Sox2*. We added the results in Figure 1D and Figure 1 —figure supplement 2A.

2. The evidence that the developmental over-proliferation of the Tsc1 cko progenitors and the Muller glial degeneration are linked is partly supported by the fact that deletion of Tsc1 in mature Muller glia did not cause this phenotype. However, it is not clear that the Tsc1 was effectively deleted in this experiment. More evidence to show this experiment actually produced the deletion should be provided.

The best way to determine the deletion of Tsc1 gene in Glast-CreER-affected MG might be examining Tsc1 expression in R26tdTom-positive cells by co-immunostaining. However, the anti-Tsc1 antibody did not work for immunostaining. Thus, instead, we detected Cre-dependent deletion in Tsc1 gene locus by PCR. Our results show that the floxed DNA, which includes exon 17 and 18 of mouse Tsc1 gene, was deleted only in Tsc1(f/f);Glast-CreER mouse retina injected with tamoxifen. The results suggest that Tsc1 was deleted in the Glast-CreER-affected mouse MG. The efficacy of Glast-CreER has also been confirmed in many previous reports, including the original paper (de Melo et al., 2012).

**Author response image 7. sa2fig7:** PCR detection of Glast-CreER-mediated deletion in mouse Tsc1 gene locus. (A) Schematic diagram of Tsc1 gene locus of wt and Tsc1^flox^ mice. (B) Agarose gel images of DNA bands amplified by the PCR with indicated primers. 1. loxP containing DNA fragments of Tsc1 gene, which were not affected by Cre recombinase (495bp). 2. wt DNA fragments of Tsc1 gene (295bp). 3. 2.01kbp and 2.24kbp DNA fragments are expected to be amplified from wt and Tsc1^flox^ alleles, respectively. However, those are too big to be amplified in our PCR condition (extension time = 60 seconds). 4. Tsc1 exon 17&18 were deleted in MG subpopulation by Glast-CreER^t2^ only upon Tam injection (368bp). 5. Tsc1 exon 17&18 were deleted in majority retinal cells by Chx10-Cre (368bp).

3. One of the most interesting aspects of the report is that only the deletion of Tsc1 using the Trp1-cre line leads to the phenotypes. One interpretation of this is that the hyper-proliferation of the Tsc1 cko progenitors only occurs when there are normal cells nearby. The α-Pax6cre line shows a reduced phenotype, consistent with this idea, while when Tsc1 is knocked out in all progenitors across the retina, proliferation appears to be normal. They interpret these different phenotypes in terms of the "age" of the progenitors or the number of total cell divisions, but it seems more consistent with a model where normal and Tsc1 deficient progenitors compete for space and the Tsc1 deficient cells outcompete normal cells, but not other Tsc1 deficient progenitors. This alternative model could be tested by looking more closely at the percentages of EdU+ cells near the boundary of the conditional deletion. Probably best to do this early in the α-Pax6cre line where the boundary is likely to be sharp.

We agree that the space allowance for a Tsc1-deficient RPC is also a critical factor that determines whether it can divide exceeding the division limit. To be a dominant clone in a space, a fast-dividing Tsc1-deficient RPC should have majority WT RPCs that divide slowly in its neighbor, as the case of Tsc1(f/f);TRP1-Cre mice. However, the neighbors of a Tsc1-deficient RPC are also mostly Tsc1-defcient RPCs in the other Tsc1-cko mice, thus the RPC cannot hyperexpand exceeding its neighbors.

We also compared EdU(+) cells in tdTom(+) Tsc1-deficient clones and adjacent tdTom(-) WT clones in P0 Tsc1(f/f);TRP1-Cre mouse retinas. We could find the increase of EdU-positivity in tdTom(+) clones in the mouse retina, suggesting the hyperproliferation of Tsc1-deficient RPCs over neighboring WT RPCs (please see the results Author response image 8). In our previous report of Tsc1-cko by Chx10-Cre (Choi et al., 2018), we had also compared the proliferation rate of R26(+) potential Tsc1-deficient cells and R26(-) WT cells in the same retina. The results showed enhanced cell proliferation in R26(+) areas in comparison to their adjacent R26(-) areas.

**Author response image 8. sa2fig8:** Relative overproliferation of Tsc1-deficient RPCs in Tsc1^f/f^;TRP1-Cre mouse retina. (A) EdU-labeled proliferating cells and those moved to G2/M phase of cell cycle to express pH3 in P0 mouse retinas were examined by co-immunostaining after the injection of EdU to the mice at 3 h prior to tissue preparation. (B) Numbers of EdU-labeled proliferating cells in R26tdTom(+) Cre-affected cell areas and R26tdTom(-) wild-type cell areas were counted and shown in the graph. (C) Numbers of EdU;pH3-positive cells in R26tdTom(+) Cre-affected areas and R26tdTom(-) wild-type cell areas were counted and shown in the graph.

4. The neurosphere assay is not very quantitative to show differences between the Tsc1 cko (Figure 4), particularly when they do not see differences until the spheres are grown for 4 weeks, long past when proliferation would normally have finished in the retina. The in vivo cell cycle analyses in Figure 7 do a better job anyway.5. They rescue the rosetting phenotype by crossing the Tsc1 cko mice with a Hif1a deletion. The "hyper-expansion" phenotype is less well rescued (Figures 6B, 6D). Thus it seems like these two phenotypes might be un-related. The authors should discuss this possibility.

mTORC1 might regulate cell growth and proliferation via multiple downstream targets. Hif1alpha is one of the targets, therefore Hif1alpha deletion could not normalize completely the Tsc1-cko phenotypes. Besides Hif1alpha, the hyperproliferation of Tsc1-deficient RPCs might be mediated by multiple mTORC1 targets including the immunoproteasome, of which loss was also insufficient to rescue Tsc1-cko phenotypes (Figure 6 —figure supplement 1). However, Hif1a deletion is enough to reduce the Tsc1-deficient clone size below the threshold level, which is necessary for the degeneration of MG.

6. The authors argue that Muller glia undergo senescence because they are the last cell type generated and the progenitors have exceeded their mitotic limit. This is not exactly true. Muller glia are among the last cells generated, but rods and bipolar cells are also included in the last cell divisions. However, these other last generated cells do not seem to be undergoing cell death or senescence-related gene expression. This is specifically demonstrated in supplemental Figure S9. Rather it appears that there is a specific requirement for Tsc1 in Muller glia, perhaps near the end of neurogenesis, much like there is for p27kip. It would be interesting to determine whether the premature end of neurogenesis might lead to incomplete differentiation of the Muller glia.

Thank you for the comment that provides us an alternative interpretation of the phenotype. In this study, we have not determined whether MG cells differentiate completely or not. Only mature MG marker we examined in this study is GS, which was expressed properly in Tsc1(f/f);TRP1-Cre as well as Tsc1(f/+);TRP1-Cre mice (Figure 1C). The GS-positive cells were even produced at higher number in P7 Tsc1(f/f);TRP1-Cre mouse retinas in comparison to those in Tsc1(f/+);TRP1-Cre littermate retinas (Figure 1C; Figure 1 —figure supplement 2B). However, the correct answer for the question could be provided by comprehensive analyses of gene expression in MG of the mouse retinas. The analyses may need scRNA-seq or RNA-seq of purified MG in the mouse retinas. Thus, it should be done in a separate study in future.

[Editors' note: further revisions were suggested prior to acceptance, as described below.]

Essential revisions:1) Please discuss the potential reasons as of why Chx10-Cre null neurospheres divide better in vitro as indicated by reviewer 1.

We think that our explanation for the conclusion might not be enough in the previous versions. Our conclusion that Tsc1-deficient RPCs can have the highest over proliferation potential when they are in minority (i.e., the deletion by Tyrp1-Cre) does not emphasize the importance of non-autonomous factors provided by neighboring wild-type cells. A space that is not occupied by wild-type clones can be the factor, if there is a factor contributed by the neighboring wild-type cells.

As the reviewer indicates, our neurosphere results show the exhaustive over proliferation can also happen to Chx10-Cre Tsc1-deficient RPCs (Figure 5F – 5H; Figure 5 —figure supplement 3F), which did not expand exceeding the limit in vivo (Figure 5A and 5C). One important difference of the RPCs in the neurosphere culture from those in vivo is a space to expand their clones. The space allowed for a Chx10-Cre Tsc1-deficient RPC in the neurosphere culture is enough to expand until they reach division limit, whereas the space is limited in mouse retinas. Furthermore, the Tsc1-deficient RPC has to compete with another Tsc1-deficient RPCs (>80% of neighbors) for a retinal space in Tsc1(fl/fl);Chx10-Cre mouse retinas. This chance to have another Tsc1-deficient RPCs in the neighbor is much lower in Tsc1(fl/fl);Tyrp1-Cre mouse retina (<50% of neighbors). Therefore, a Tsc1-deficient RPC could expand more freely in Tsc1(fl/fl);Tyrp1-Cre mouse retina than the other three Tsc1-cko mouse retinas (i.e., Rax-Cre, Chx10-Cre, and α-Cre).

We modified the conclusion by reflecting these points (page 9 and 21). To help the readers’ understanding the data, we also provide a hypothetical model diagram in Figure 4 —figure supplement 4.

2) Please discuss why Tcs1 CKOs MG (based on tdTomato Cre reporter expression) seem to be unable of undergoing reactive gliosis, as pointed out by reviewer 2.

Currently, we cannot provide the correct answer why the MG derived from Tsc1-deficient RPCs do not exhibit the characteristics of reactive gliosis. The Tsc1(fl/fl);Tyrp1-Cre mouse retina exhibited the increases of Iba1-positive microglia cell number and the hyper-extension of astrocyte processes (please see the results in the following response to the reviewer’s comment), indicating that the damages are present in the retina. These results also suggest that mTORC1 hyperactivation in Tsc1-deficient MG might suppress the gliotic responses. The mechanism for this possibility, however, should be investigated in a future study.

3) Please change the statement on page 21 that "The MG are the last-born retinal cell type, arising around the first postnatal week in mice." As reviewer 3 indicate this is not correct. Also discuss why rod and bipolar that are generated around the same time as Müller glia do not undergo the same apoptosis.

We appreciate for the correction. We corrected it in the revised manuscript (highlighted in page 21).

4) Please also discuss possible peculiarities of the ciliary margin, as indicated by reviewer 3.

In the revised Discussion (page 22), we added our interpretation why the effects of Tsc1 deletion could be greater in the CM RPCs than the central RPCs.

Reviewer #1:The authors have added important new experiments. First, they showed that using a TRP1-Cre-ER system, in which they activate Cre at E9.5 with tamoxifen, they obtain similar results to that observed with the TRP1-Cre system. This result addresses the concerns of the other reviewers that the integration site of TRP1-Cre may have been contributing to the phenotype.Second, they isolated E13 progenitors from mice in which Tsc1 was deleted using Chx10-Cre, which does not induce the over-proliferation/Muller glia degeneration phenotype in vivo, and assessed resultant neurosphere size and number. They compared Tsc1 null to Tsc1 heterozygous neurospheres. Interestingly, Tsc1 null progenitors generated ~2-fold more neurospheres, and the spheres were somewhat (~20%) larger than those from Tsc1 hets. They then degenerated, mimicking the effect seen in vivo when TRP1-Cre is used to knockout Tsc1. These data imply that the effects of the knockout are cell autonomous, and are not a consequence of cell competition.I commend the authors for their extra work to address the concerns of the reviewers, and the paper should be published. However, I do feel there should be an addition to the Discussion. The neurosphere/Chx10-Cre result also suggests that there is something about the in vivo milieu of the mice lacking the proliferation phenotype (Chx10-Cre, Rx-Cre etc) that suppresses the proliferative effect of Tsc1 deletion. One possibility, raised in the author's rebuttal, is that the ratio of mutant to normal cells must be low for the proliferative defect to occur, implying either that normal cells stimulate the mutant cell division, or excess mutant cells inhibit over-proliferation.

We think that our explanation for the conclusion might not be enough in the previous versions. Our conclusion does not emphasize the importance of non-autonomous factors provided by majority neighboring wild-type cells for the over proliferation of Tsc1-deficient RPCs. A space that is not occupied by wild-type clones can be the factor, if there is a factor contributed by the neighboring wild-type cells. Please see the details in our response below.

Against the idea that normal cells stimulate over-proliferation, they now show that Chx10-Cre null neurospheres divide better in vitro. That leaves the notion that high numbers of Tsc1 cells may inhibit their own proliferation. The Discussion is completely lacking any mention of this confusing and unsolved aspect of the data, nor does it even acknowledge the paradox. At the very least the authors could highlight the problem and state that the difference in the effects of various Cre models in vivo coupled with the neurosphere data in vitro is puzzling, and the mechanism as to why mutant progenitors must be present at a low frequency to over-proliferate is unresolved.

As we show in vitro neurosphere culture (Figure 4, F – H and figure supplement 3) and EdU-labeling in vivo (Figure 7, A – C), Tsc1-deficient RPCs are intrinsically capable of over proliferation regardless of the Cre drivers. The exhaustive proliferation, however, can occur only if the spatial limitation is not present.

As we show in the hypothetical model diagram in Figure 4 —figure supplement 4, by having more wild-type RPCs in its neighbor, a Tsc1-deficient RPC in Tsc1(fl/fl);Tyrp1-Cre mouse retina can have a higher chance than that in Tsc1(fl/fl);Chx10-Cre mouse retinas to divide extra rounds and fill a free space, which neighboring wild-type RPCs did not fill yet. This invasive clonal expansion will continue until the Tsc1-deficient RPCs reach the division limit and cannot divide more. Based on these, we concluded that Tsc1-deficient RPC population should be lower than a threshold level that a Tsc1-deficient RPC clone can escape the competition with another Tsc1-deficient RPC and over proliferate exceeding the division limit.

This type of competitive clonal expansion could also occur in other conditions, where two RPC populations expand at different speeds. The Rptor(fl/fl);Tyrp1-Cre mouse retina, which is composed of relatively fast dividing wild-type RPCs and slowly dividing (or cell cycle arrested) Rptor-deficient RPCs, could be also one of the examples. However, wild-type RPCs in the Rptor(fl/fl);Tyrp1-Cre mouse retina could not over proliferate even though their neighboring Rptor-deficient RPCs stopped expanding (Figure 5F – 5J). The results suggest that the over proliferation is not a relative property obtained by the division differences but a specific property of Tsc1-deficient RPC.

Combining these, we modified our interpretation for the clonal hyper-expansion in Discussion (second paragraph in page 21). We also provide a model diagram that shows the hypothetical RPC expansion in each Tsc1-cko and Rptor-cko mouse retinas (Figure 4 —figure supplement 4).

Reviewer #2:The reviewers have satisfied most of my concerns.However, the mutant GFAP immunofluorescence raises an interesting question. The authors clearly show the presence of residual SOX9+ MGs within the mutant retinae with lamination defects. However, these mutant retinae, which are clearly experiencing damage, do not upregulate GFAP. These data suggest that these presumably Tcs1 CKOs MG (based on tdTomato Cre reporter expression) are not capable of undergoing reactive gliosis. Is MG differentiation, or the ability to detect damage, or Gfap transcription compromised?

(1) Regarding to MG differentiation: Given the expression of GS (Figure 1C; Figure 1 —figure supplement 2B), a marker for mature MG, we believe MG are likely differentiated properly in the Tsc1(fl/fl);Tyrp1-Cre mouse retina.

(2) Regarding to the retinal damage: Tsc1(fl/fl);Tyrp1-Cre mouse retina exhibited the increase of Iba1-positive microglia cell number and the hyper-extension of astrocyte processes into the retina (please see the results in Author response image 9). These results suggest the presence of the damages in the retina. However, we do not know whether the damage sensing mechanism of Tsc1-deficient MG is compromised or not.

**Author response image 9. sa2fig9:** Distribution of glia in the mouse retina.

(3) Regarding to Gfap expression: It would be possible that hyperactive mTORC1 suppresses the expression of Gfap in the MG. However, we do not know whether Gfap transcription is compromise until we assess the transcription efficiency at the Gfap locus. We would like to leave this for a future work.

Reviewer #3:The authors have done a very good job at addressing my previous concerns. However, I think the text needs a few changes for accuracy. The authors state on page 21 that "The Mg are the last-born retinal cell type, arising around the first postnatal week in mice." This is not correct. The MG are AMONG the last cells born in the retina, but rods and bipolar cells are included in two cell clones that contain Muller glia. Therefore, the authors should change this sentence accordingly.

Thank you for the correction. We corrected and highlighted it in the manuscript (page 21).

Moreover, since at least some of the rods and most of the bipolar cells have been generated in the KO mice by progenitors that have undergone as many rounds of division as those that generated the MG, one might presume their survival would be similarly affected. Yet they appear not to undergo the apoptosis that the MG are subject to in these mice. The authors should discuss this point in the Discussion.

MG might be in less terminal stage than the other last-born types (i.e., rPR and BC) in terms of differentiation, since they can resume cell cycle to regenerate neurons in the injured cold-blooded vertebrate retinas (reviewed by Lahne et al., 2020). Mouse MG could also divide after the injury, if histone deacetylase (Hdac) inhibitor is provided after viral expression of a proneural transcription factor achaete-scute homolog 1 (Ascl1) (Jorstad et al., 2017). Furthermore, given the fact that developing cells are more sensitive than terminally differentiated cells to cell death (Fuchs and Steller, 2011; Vaux and Korsmeyer, 1999), MG therefore are likely more sensitive to the cell death than rPR and BC. We added this interpretation with the references in Discussion (page 22 and 23).

The second point that could bear some elaboration is the possibility that there is something unique about the Trp-cre expressing cells at the retinal margin. The authors show in Figure 4 that there is a progressive decline in RPC overgrowth from peripheral to central retina when the cre is targeted to progressively more RPCs, but these different cre lines also have a relative decrease in the contribution of RPCs adjacent to the CM. It is possible that this phenotype is primarily due to expansion in the proliferation of a relatively small subset of RPCs, ones that may more closely resemble the CMZ cells of lower vertebrates. The authors should consider/discuss this possibility.

We agree that the deletion of Tsc1 in a small RPC subset, like those in the CM, leads to the phenotypes through the competitive over proliferation of the Tsc1-deficient RPCs against majority wild-type RPCs. The CM RPCs were found to proliferate less robustly than those in the central retina in mouse embryo (Bélanger et al., 2017; Marcucci et al., 2016), so did the RPCs in lower vertebrate CMZ (Harris and Perron, 1998). Therefore, the effects of Tsc1 deletion that accelerates RPC cell cycle might be greater in the CM RPCs than the majority central RPCs. We added this interpretation with the references in Discussion (page 21).